# A host defense peptide mimetic, brilacidin, potentiates caspofungin antifungal activity against human pathogenic fungi

Thaila Fernanda dos Reis[1], Patrícia Alves de Castro[1], Rafael Wesley Bastos[1], Camila Figueiredo Pinzan[1], Pedro F. N. Souza [2], Suzanne Ackloo [3], Mohammad Anwar Hossain [4], David Harold Drewry [4,5], Sondus Alkhazraji[6], Ashraf S. Ibrahim [6,7], Hyunil Jo [8], Jorge D. Lightfoot[9], Emily M. Adams [9], Kevin K. Fuller[9,10], William F. deGrado[8] & Gustavo H. Goldman [1] ✉

Fungal infections cause more than 1.5 million deaths a year. Due to emerging antifungal drug resistance, novel strategies are urgently needed to combat life-threatening fungal diseases. Here, we identify the host defense peptide mimetic, brilacidin (BRI) as a synergizer with caspofungin (CAS) against CAS-sensitive and CAS-resistant isolates of *Aspergillus fumigatus, Candida albicans, C. auris*, and CAS-intrinsically resistant *Cryptococcus neoformans*. BRI also potentiates azoles against *A. fumigatus* and several Mucorales fungi. BRI acts in *A. fumigatus* by affecting cell wall integrity pathway and cell membrane potential. BRI combined with CAS significantly clears *A. fumigatus* lung infection in an immunosuppressed murine model of invasive pulmonary aspergillosis. BRI alone also decreases *A. fumigatus* fungal burden and ablates disease development in a murine model of fungal keratitis. Our results indicate that combinations of BRI and antifungal drugs in clinical use are likely to improve the treatment outcome of aspergillosis and other fungal infections.

Fungal diseases occur in >1 billion people worldwide and are responsible for 1.5 million deaths a year[1,2]. Aspergillosis encompasses a group of heterogeneous diseases caused by *Aspergillus* spp[3]. In immunocompetent and immunosuppressed patients, aspergillosis is characterized by noninvasive and invasive diseases, respectively[3–7]. The most lethal form of aspergillosis in recipients of both hematopoietic stem cells and solid-organ transplants is invasive pulmonary aspergillosis (IPA) and *A. fumigatus* is the leading cause of this disease, which comprises >300,000 cases worldwide and is associated with a mortality rate of up to 90% in the most susceptible populations[2,3,8–11]. In addition, *Aspergillus* sp. are an important source of fungal keratitis (FK), a threating infection of the cornea that impacts an estimated 1–2 million people annually worldwide[12,13]. The incidence of FK is highest in tropical countries with high agricultural activity, where infection frequently results from corneal abrasion by vegetative debris[12,13]. In the United States

[1]Faculdade de Ciências Farmacêuticas de Ribeirão Preto, Universidade de São Paulo, Ribeirão Preto, Brazil. [2]Visiting professor at Drug Research and Development Center, Department of Physiology and Pharmacology, Federal University of Ceará, Fortaleza, Ceará 60451, Brazil. [3]Structural Genomics Consortium, University of Toronto, 101 College Street, MaRS South Tower, Suite 700, Toronto, ON M5G 1L7, Canada. [4]Structural Genomics Consortium, UNC Eshelman School of Pharmacy, University of North Carolina at Chapel Hill, Chapel Hill, NC 27599, USA. [5]Lineberger Comprehensive Cancer Center, Department of Medicine, School of Medicine, University of North Carolina at Chapel Hill, Chapel Hill, NC 27599, USA. [6]Division of Infectious Diseases, The Lundquist Institute for Biomedical Innovation at Harbor-University of California Los Angeles (UCLA) Medical Center, Torrance, CA 90502, USA. [7]David Geffen School of Medicine at UCLA, Los Angeles, CA 90095, USA. [8]Department of Pharmaceutical Chemistry, University of California, San Francisco, San Francisco, CA 94143, USA. [9]Department of Ophthalmology, University of Oklahoma Health Sciences Center, Oklahoma City, OK 73104, USA. [10]Department of Microbiology and Immunology, University of Oklahoma Health Sciences Center, Oklahoma City, OK 73104, USA. ✉e-mail: ggoldman@usp.br

and other developed countries, contact lens wear has emerged as the predominant risk factor[12,13]. Azoles (itraconazole, posaconazole, voriconazole, and isavuconazole) are fungicidal drugs for *A. fumigatus* and are used as first-line therapy against IPA. While the fungistatic echinocandins, such as caspofungin (CAS), can be used as salvage therapy and have been recommended in combination therapies against emerging azole-resistant infections[14,15]. Azoles inhibit the ergosterol biosynthesis pathway by directly targeting the *cyp51/erg11* encoding the lanosterol 14-demethylase[16,17]. CAS blocks fungal cell wall synthesis by noncompetitively inhibiting the fungal β−1,3-glucan synthase (Fks1), required for the biosynthesis of β−1,3-glucan[18]. Specifically, caspofungin affects the composition and organization of the *A. fumigatus* cell wall with hyphae hyper-branching, lysis of hyphal apical compartments, resulting in loss of cell wall β−1,3-glucan. In the presence of CAS, the cell wall integrity (CWI) pathway is activated and the CWI mitogen-activated protein kinase MpkA and the phosphatase calcineurin turn on the transcription factors RlmA and CrzA, which translocate to the nucleus and regulate the activation of several stress responses and cell wall modifications, including chitin synthase gene expression resulting in overproduction of chitin[19,20].

The increasing number of azole-resistant environmental isolates due to the overuse of this class of antifungal drugs in agriculture, clinical azole-resistant *A. fumigatus* isolates are currently a crucial problem and a major threat to immunosuppressed patients[21–27]. Because of the scarcity in antifungal agents currently in development[28], repurposing of currently approved drugs alone or in combination with currently used antifungal agents, represents a potential opportunity for the discovery of new modality of treatments[29,30]. By using this strategy, several compounds have already been identified as potential new antifungal agents, and more importantly as potentiators of antifungal drugs currently in clinical use[31–37]. For an example, an efficient approach to control aspergillosis could be the identification of compounds that can potentiate CAS to becoming fungicidal.

In this work, we screen four chemical collections containing a total of 1402 compounds aiming to identify compounds that can enhance the in vitro CAS activity against *A. fumigatus*. We identify 5 CAS enhancers (compounds that can partially inhibit the fungal growth at 20 μM and have increased inhibition when combined with CAS) and 12 CAS synergizers (compounds that cannot inhibit the fungal growth at 20 μM but have increased inhibition when combined with CAS). Among the CAS synergizers, we closely investigate a small molecule host defense peptide mimetic, brilacidin (BRI), which is in clinical development for multiple indications. BRI has previously exhibited broad-spectrum inhibitory activity in bacteria and viruses, as well as immunomodulatory/anti-inflammatory properties (Innovation Pharmaceuticals Inc., 2022, http://www.ipharminc.com/brilacidin-1). BRI is able to synergize with echinocandin and azoles activities enhancing the cidal activity of CAS, voriconazole (VOR) and posaconazole (POSA) against *A. fumigatus* and Mucorales fungi, respectively. Further, BRI is efficient against *A. fumigatus* biofilm and can overcome CAS-resistance but not VOR-resistance. BRI can also synergize with CAS against other human pathogenic fungi, such as *C. albicans*, *C. auris*, and *C. neoformans*. By a combination of screening an *A. fumigatus* phosphatase null mutant library and a collection of protein kinase inhibitors, we were able to assign the CAS synergism mechanism of action to the cell wall integrity (CWI) pathway. Both calcineurin and the CWI mitogen-activated protein kinase (MAPK) MpkA are important for the mechanism of action of BRI in increasing CAS activity. Importantly, BRI combined with CAS significantly clears *A. fumigatus* lung infection in an immunosuppressed murine model of invasive pulmonary aspergillosis. BRI alone also decreases *A. fumigatus* fungal burden and ablates disease development in a murine model of fungal keratitis.

## Results

### Identification of a host defense peptide mimetic, BRI, as a CAS synergizer

To identify compounds that can enhance or synergize with CAS activity against *A. fumigatus*, we used the Minimal Effective Concentration (MEC) assay to screen the fungus susceptibility to four chemical drug libraries: (i) the COVID Box (containing 160 compounds, see https://www.mmv.org/mmv-open/archived-projects/covid-box); (ii) the Pandemic Response Box (containing 400 compounds, see https://www.mmv.org/mmv-open/pandemic-response-box/about-pandemic-response-box); (iii) the National Institutes of Health (NIH) clinical collection (NCC) (containing 727 compounds; see https://pubchem.ncbi.nlm.nih.gov/source/NIH%20Clinical%20Collection); and (iv) the epigenetic probe library (containing 115 compounds, see https://www.sgc-ffm.uni-frankfurt.de/). In total, two methods were employed to assess 1402 compounds by using a combination of 0.2 μg/mL of CAS (a concentration below the MEC; MEC of CAS is 0.25 μg/mL) and up to 20 μM of each compound compared to the effect on growth of *A. fumigatus* of each drug alone. First, we assessed growth by two independent rounds of visual inspection and selected 17 compounds that could inhibit *A. fumigatus* growth. Second, *A. fumigatus* growth in the presence of CAS 0.2 μg/mL alone, each of these 17 compounds at 20 μM alone, and a combination of each of these compounds from 0.6 to 20 μM plus CAS 0.2 μg/mL was quantified by using Alamar blue (Fig. 1a). Based on this assay we defined enhancers as the compounds that alone could inhibit over 30% of *A. fumigatus* metabolic activity but in combination with CAS inhibited even more, while synergizers were defined as compounds which alone inhibited <30% of the fungal metabolic activity but in combination with CAS inhibited >30% (for further definitions and applications of synergizers and enhancers, see[38–41].). Five compounds were classified as enhancers (Fig. 1a): (i) chlormidazole (https://go.drugbank.com/drugs/DB13611) and (ii) ravuconazole (https://go.drugbank.com/drugs/DB06440; both inhibiting ergosterol biosynthesis), (iii) 5-fluorocytosin (https://go.drugbank.com/unearth/q?c=_score&d=down&query=5-fluorocytosin&searcher=drugs; that inhibits the RNA and DNA biosynthesis), (iv) ciclopirox (https://go.drugbank.com/unearth/q?utf8=%E2%9C%93&searcher=drugs&query=ciclopyxox); it is thought to act through the chelation of $Fe^{3+}$ and $Al^{3+}$, polyvalent metal cations[42], and (v) MMV1593544 (a possible antiviral compound that inhibits SARS-CoV-2 infection in vitro[43]). Twelve compounds were classified as synergizers (Fig. 1a, b): (i) toremifene (https://go.drugbank.com/drugs/DB00539; a nonsteroidal triphenylethylene derivative used as an antitumor drug that appears to bind to the estrogen receptors competing with estradiol), (ii) brilacidin (https://go.drugbank.com/drugs/DB12997); a compound that acts as a mimetic of host defense peptides[44], (iii) MMV1634399 (a quinoline anti-malarial)[45], (iv) Diiodoemodin or MMV1581545 (an anti-bacterial emodin derivative)[46], (v) PPTN (a potent, high-affinity, competitive and highly selective nucleotide-sugar-activated P2Y14 receptor antagonist)[47], (vi) triclopidine (a prodrug that is metabolized to an active form, which blocks the ADP receptor that is involved in GPIIb/IIIa receptor activation leading to platelet aggregation; https://go.drugbank.com/drugs/DB00208), (vii) loxoprofen (a non-steroidal anti-inflammatory drug that acts as a non-selective inhibitor of cyclooxygenase enzymes, which are responsible for the formation of various biologically active pain, fever, and inflammatory mediators; https://go.drugbank.com/drugs/DB09212), (viii) regorafenib (a small molecule inhibitor of multiple membrane-bound and intracellular kinases involved in normal cellular functions and in pathologic processes such as oncogenesis, tumor angiogenesis, and maintenance of the tumor microenvironment; https://go.drugbank.com/drugs/DB08896), (ix) OSU 03012 (a potent inhibitor of recombinant phosphoinositide-dependent kinase 1)[48], (x) MMV1782211 (an inhibitor of the SARS-CoV-2 main protease);

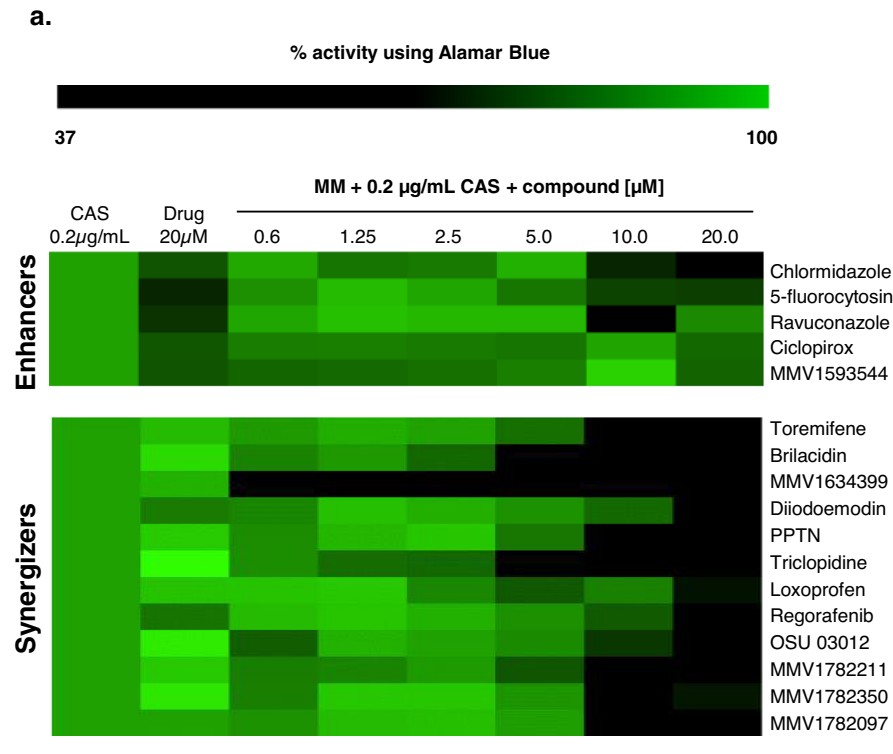

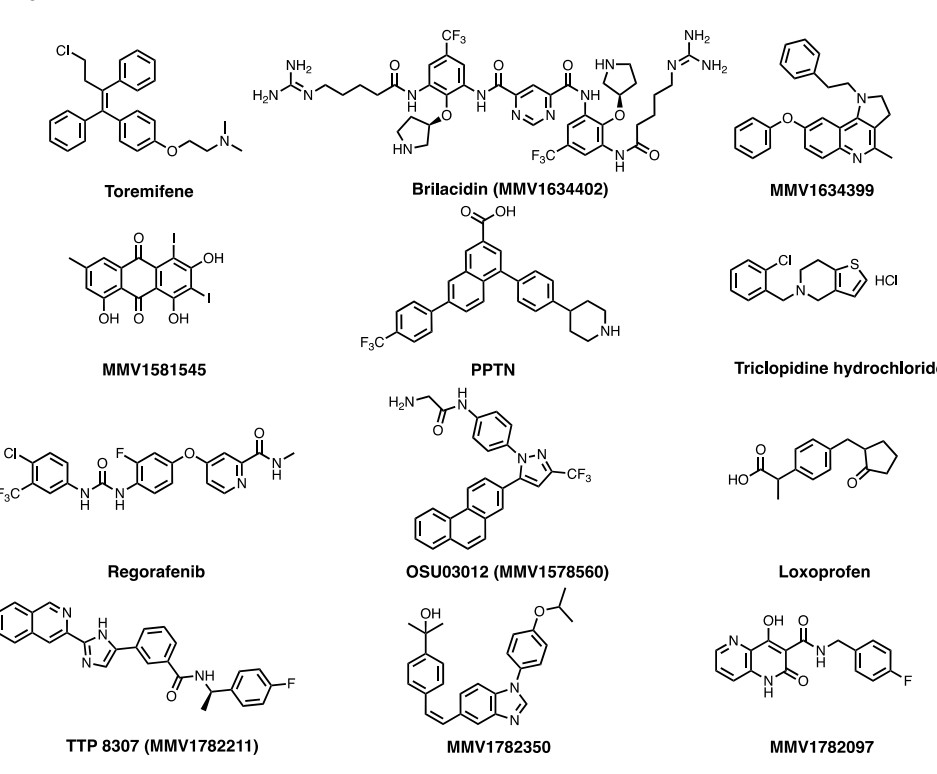

**Fig. 1 | Screening of repurposing chemical libraries identify several compounds that enhance or synergize caspofungin activity. a** Heat map of % of metabolic activity using Alamar blue. The % of activity is based on *A. fumigatus* grown for 48 hs at 37 °C in the absence or presence of a specific compound (MM + CAS [0.2 μg/mL] or MM + enhancers or synergizers alone [20 μM], or a combination of MM containing CAS [0.2 μg/mL]) and added of enhancer or synergizers [0.6–20 μM] divided by the control (MM), both grown for 48 h at 37 °C. The results are the average of three independent experiments analyzed in technical duplicates each. Heat map scale and gene identities are shown. **b** Chemical structures of the CAS synergizers.

**Table 1 | BRI+CAS can overcome *A. fumigatus* CAS-resistance**

| Strain | MEC CAS [µg/mL] | MIC BRI [µM] | CAS [µg/mL] | | | BRI [µM] | | 0.25 µg/mL CAS | | 2 µg/mL CAS | | 4 µg/mL CAS | |
|---|---|---|---|---|---|---|---|---|---|---|---|---|---|
| | | | 0.25 | 2 | 4 | 20 | 40 | 20 µM BRI | 40 µM BRI | 20 µM BRI | 40 µM BRI | 20 µM BRI | 40 µM BRI |
| WT | 0.25 | >80 | + | + | + | + | + | – | – | – | --- | – | – |
| CM7555 CAS^R | 16 | >80 | + | + | + | + | + | + | – | + | – | + | – |
| DPL1033 CAS^R | 16 | >80 | + | + | + | + | + | + | – | + | – | + | – |
| MD24053 CAS^R | 16 | >80 | + | + | + | + | + | + | – | + | – | + | – |
| CYP-15-184 | 0.25 | >80 | + | ND | ND | + | ND | – | ND | ND | ND | ND | ND |
| CYP-15-190 | 0.25 | >80 | + | ND | ND | + | ND | – | ND | ND | ND | ND | ND |
| CYP-15-192 | 0.25 | >80 | + | ND | ND | + | ND | – | ND | ND | ND | ND | ND |
| CYP-15-195 | 0.25 | >80 | + | ND | ND | + | ND | – | ND | ND | ND | ND | ND |
| CYP-15-202 | 0.25 | >80 | + | ND | ND | + | ND | – | ND | ND | ND | ND | ND |
| CYP-15-212 | 0.25 | >80 | + | ND | ND | + | ND | – | ND | ND | ND | ND | ND |
| CYP-15-213 | 0.25 | >80 | + | ND | ND | – | ND | – | ND | ND | ND | ND | ND |
| CYP-15-215 | 0.25 | >80 | + | ND | ND | – | ND | – | ND | ND | ND | ND | ND |
| CYP-15-220 | 0.25 | >80 | + | ND | ND | + | ND | – | ND | ND | ND | ND | ND |
| CYP-15-221 | 0.25 | >80 | + | ND | ND | + | ND | – | ND | ND | ND | ND | ND |
| CYP-15-222 | 0.25 | >80 | + | ND | ND | + | ND | – | ND | ND | ND | ND | ND |
| CYP-15-224 | 0.25 | >80 | + | ND | ND | + | ND | – | ND | ND | ND | ND | ND |
| CYP-15-225 | 0.25 | >80 | + | ND | ND | + | ND | – | ND | ND | ND | ND | ND |
| CYP-15-226 | 0.25 | >80 | + | ND | ND | + | ND | – | ND | ND | ND | ND | ND |
| CYP-15-228 | 0.25 | >80 | + | ND | ND | + | ND | – | ND | ND | ND | ND | ND |
| CYP-15-229 | 0.25 | >80 | + | ND | ND | + | ND | – | ND | ND | ND | ND | ND |
| CYP-15-230 | 0.25 | >80 | + | ND | ND | + | ND | – | ND | ND | ND | ND | ND |
| CYP-15-231 | 0.25 | >80 | + | ND | ND | + | ND | – | ND | ND | ND | ND | ND |
| CYP-15-75 | 0.25 | >80 | + | ND | ND | + | ND | – | ND | ND | ND | ND | ND |
| CYP-15-108 | 0.25 | >80 | + | ND | ND | + | ND | – | ND | ND | ND | ND | ND |
| CYP-15-109 | 0.25 | >80 | + | ND | ND | + | ND | – | ND | ND | ND | ND | ND |
| CYP-15-147 | 0.25 | >80 | + | ND | ND | + | ND | – | ND | ND | ND | ND | ND |
| F14946 | 0.25 | >80 | + | ND | ND | + | ND | – | ND | ND | ND | ND | ND |
| 20089320 | 0.25 | >80 | + | ND | ND | + | ND | – | ND | ND | ND | ND | ND |

Obs.: *CAS* caspofungin, *BRI* Brilacidin. CM7555, DPL1033, and MD24053 are clinical isolates resistant to caspofungin. The mechanism of CAS^R in CM7555 is unknown while strains DPL1033 and MD24053 have mutations S679P and F675S in the *fks1* gene. (+): CAS Minimal Effective Concentration (MEC) inhibition; (–), more inhibition than CAS MEC inhibition; and (–), total inhibition; *ND* not determined.

(https://chemrxiv.org/engage/chemrxiv-article-details/60c753ed469 df403bef44e65), (xi) MMV1782350 and (xii) MMV1782097 (two uncharacterized antivirals).

Taken together, these results suggest we were able to identify many compounds that can enhance or synergize the activity of CAS against *A. fumigatus*. The synergizers have very different mechanisms of action and targets, most of them apparently not conserved in *A. fumigatus*.

**The BRI+CAS combination have synergism against *A. fumigatus* and overcomes CAS-resistance**

Although other compounds, like MMV1634399, showed the highest potentiation of CAS activity, we decided to concentrate our further analysis on the host defense peptide mimetic brilacidin (BRI) since this drug candidate has undergone several clinical trials showing therapeutic benefit with acceptable safety profiles. BRI is a small molecule, new chemical entity, created to mimic the amphiphilic structure of host defense proteins (HDPs), having one surface with positively charged groups (cationic) and the opposite surface consisting of hydrophobic groups[49–52]. With this general synthetic form, there is no need for an agent to be of the size or composition of naturally-occurring proteins to effectively function as a HDP, as the ability to act as a HDP is retained by the much smaller synthetic amphiphilic molecule. BRI MIC for *A. fumigatus* was measured as >80 µM (Table 1 and Supplementary Table S1) and the combination of 20 µM BRI with 0.2 or 0.5 µg/mL of CAS or 0.125 and 0.25 µg/mL of VOR for 48 h at 37 °C completely inhibited conidial germination while these concentrations of CAS and VOR alone allowed slow conidial germination (Fig. 2a, b). To test if these conidia were still viable after treatment, we plated them in minimal medium and tested for conidial viability assessing the number of colony forming units. The combination of 0.2 or 0.5 µg/mL of CAS with 20 µM BRI reduced *A. fumigatus* conidial viability by 85% and 100%, respectively (Fig. 2a). BRI 20 µM also synergized with subinhibitory VOR concentrations of 0.125 and 0.25 µg/mL by reducing the *A. fumigatus* conidial viability by 92% and 99%, respectively (Fig. 2a, b). The Fractional Inhibitory Concentration (FIC) index for BRI + CAS was 0.39 indicating a synergistic effect while for BRI + VOR FIC index was 1.00 indicating an additive effect against *A. fumigatus* (Fig. 2c, d).

Antimicrobial peptides target directly or indirectly the microorganism plasma membrane disrupting their membrane potential[53,54],

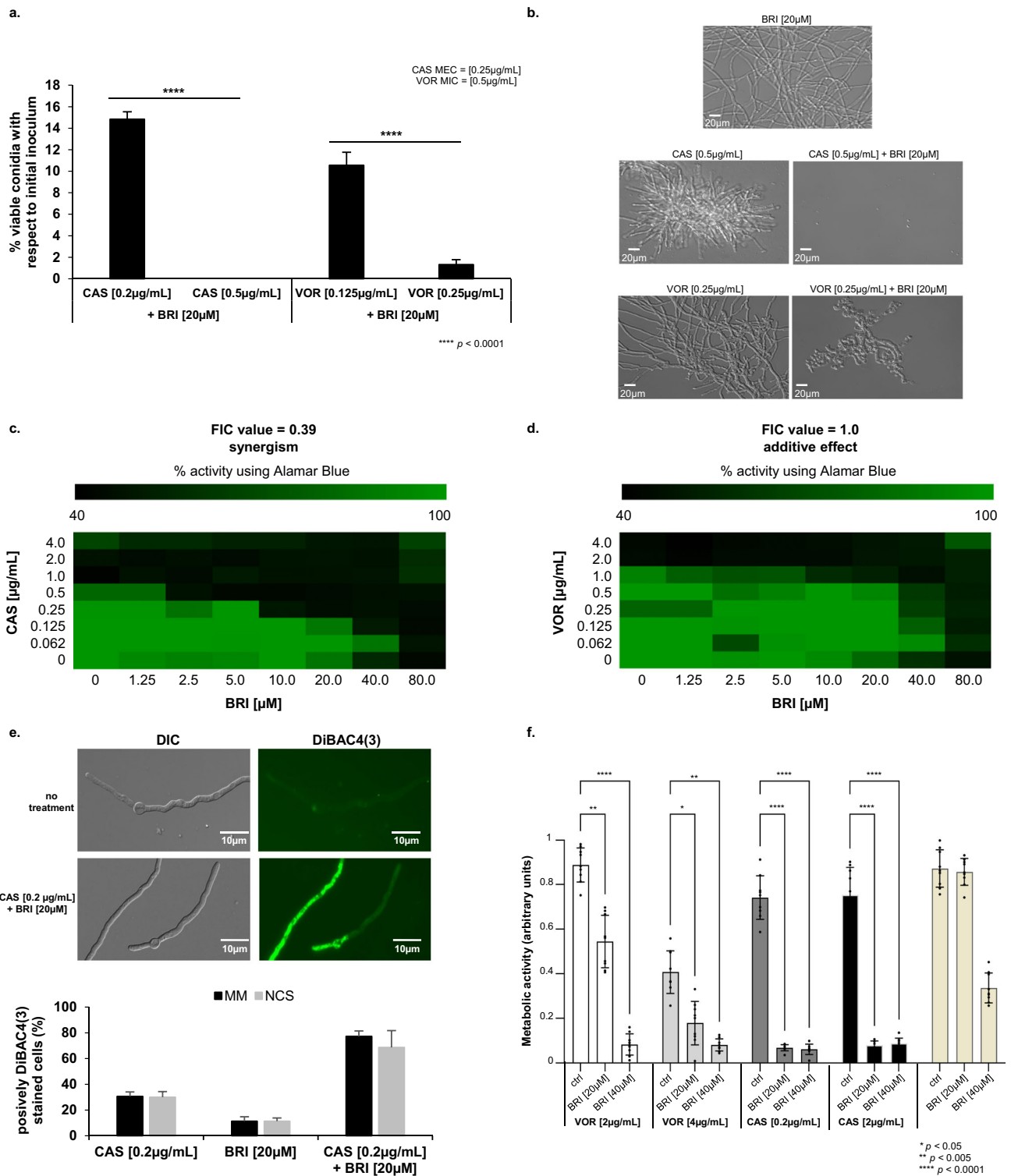

and BRI has been shown to act by a similar mechanism in various (non-fungal) microorganisms[44,55,56]. We determined the effect of BRI + CAS on the resting membrane potential by using the fluorescent voltage reporter $DIBAC_4(3)$ (increase in the fluorescent intensity indicates membrane depolarization; Fig. 2e). Untreated *A. fumigatus* had 0% fluorescent germlings while exposure to either 0.125 μg/mL CAS, 20 μM BRI, or a combination of both resulted in ~25%, 10%, and 80% fluorescent germlings, respectively (Fig. 2e). Germlings previously transferred to minimal medium (MM) without glucose (non-carbon source, NCS) for 4 h before adding CAS, BRI, or CAS + BRI for 10 min

and subsequently $DIBAC_4(3)$ have not shown any fluorescent difference with the cells that were grown in the presence of the carbon source glucose (Fig. 2e). These results suggest that BRI as well as CAS is passively transported through the cell membrane without the need for ATP.

*A. fumigatus* biofilms are more resistant to antifungal agents than planktonic cells[57]. BRI alone at 20 or 40 μM can inhibit about 10 and 60% biofilm formation, respectively (Fig. 2f) while VOR alone at 2 μg/mL or 4 μg/mL were able to inhibit about 10 and 50% biofilm formation (Fig. 2f); CAS alone at 0.25 μg/mL or 2.0 μg/mL were able to inhibit

**Fig. 2 | BRI+CAS have synergism against *A. fumigatus*. a** *A. fumigatus* conidia were incubated for 48 h at 37 °C with different combinations of BRI + CAS and BRI + VOR. After this period, non-germinated conidia were plated on MM and colony forming units (CFUs) were assessed. The results are expressed as the % of viable conidia with respect to initial inoculum (*n* = 100) and are the average of three repetitions ±SD (*t*-test; *p* < 0.0001). **b** Microscopic images of *A. fumigatus* after 48 h exposure to BRI 20 μM, CAS 0.5 μg/mL, BRI 20 μM+CAS 0.5 μg/mL, VOR 0.25 μg/mL, and BRI 20 μM+ VOR 0.25 μg/mL. Bars, 20 μm. **c, d** The Fractional Inhibitory Concentration (FIC) index for BRI + CAS and BRI + VORI, respectively. Results show the average of three independent experiments. **e** BRI + CAS disrupts the *A. fumigatus* membrane potential. *A. fumigatus* was grown 16 h at 30 °C in MM and transferred to MM without glucose (non-carbon source, NCS) or MM for 4 h before adding CAS [0.2 μg/mL], BRI [20 μM], or CAS + BRI and 3 μg/ml DIBAC$_4$(3) for 10 min. The results are expressed as the % of fluorescent cells and are the average of three independent repetitions of 50 germlings each ± SD (*n* = 50; *t*-test; *p* < 0.0001). **f** Metabolic activity expressed by XTT of *A. fumigatus* biolfilm treated with VOR or CAS alone and combinations of BRI + VOR and BRI + CAS. Biofilm was formed for 24 h of incubation at 37 °C. After this period 50 μL of fresh MM containing CAS, VOR or a combination of VOR + CAS or CAS + BRI were added to the biofilm to reach the final concentration as indicated and incubated for further 12 h at 37 °C. Untreated biofilm was used as a positive control. The graphic shows the value of 9 technical replicates from three independent experiments ± SD deviation (*n* = 3; 2way ANOVA, Tukey´s post-test; *p* < 0.05, **p* < 0.005, and ****p* < 0.0001).

about 10% biofilm formation (Fig. 2f). The different combinations BRI + VOR or BRI + CAS were able to potentiate the inhibition of the biofilm formation about 50–90% (Fig. 2f).

We also evaluated if the combination of BRI + CAS can inhibit CAS-resistant and VOR-resistant *A. fumigatus* clinical isolates (Table 1). Specifically, we tested a range of 0.25–4 μg/mL of CAS with BRI at 20 and 40 μM and VOR at concentrations of 0.5 and 2 μg/mL with BRI 20 and 40 μM. BRI had no activity against 25 *A. fumigatus* clinical isolates susceptible to CAS (MEC CAS of 0.25 μg/mL) and 3 CAS-resistant clinical strains [MEC CAS of 16 μg/mL; strains DPL1033, and MD24053 with known *fks1* mutations; and strain CM7555 with an unknown mutation(s)]. Interestingly, addition of BRI at 20 or 40 μM to CAS either partially or completely inhibited the growth of all tested strains including those that are resistant to CAS or with known resistance to azoles (Table 1). Thus, BRI clearly potentiates CAS activity against CAS- or VOR-resistant strains of *A. fumigatus*.

In contrast to the potentiation of CAS activity by BRI, addition of BRI (at 20 or 40 μM) to VOR had no effect on the resistant nature of 22 clinical isolates [15 strains with the TR34/L98H mutation and 7 strains with unknown mutation(s)] to VOR (Supplementary Table S1). In the experiment, effects were observed for 2 clinical isolates: one strain (CYP15-15-109) was totally inhibited by VOR, showing wild-type response; and, one other strain (CYP15-15-147) was partially inhibited by VOR 0.5 μg/mL+BRI but the inhibition was not seen at VOR 2.0 μg/mL+BRI (Supplementary Table S1). Curiously, the VOR-resistant clinical isolates were not inhibited by a combination of BRI + VOR but they were inhibited by BRI + CAS (compare Table 1 with Supplementary Table S1). Most of the VOR-resistant strains have increased accumulation of ergosterol since the tandem-repeat mutations at the promoter region increase the *erg11A* expression and consequently the ergosterol production[58]. Ergosterol is essential for the integrity and fluidity of fungal cell membranes and azole-induced depletion of ergosterol alters the membrane sterol composition, its stability and arrests fungal growth[59].

Taken together, these results indicate that the combination CAS + BRI can depolarize the cell membrane, decrease *A. fumigatus* biofilm formation and completely or partially overcome CAS-resistance in echinocandin-resistant *A. fumigatus* clinical isolates. VOR-resistant isolates are sensitive to CAS + BRI combinations but most of them are not sensitive to BRI + VOR combinations.

**BRI impacts *A. fumigatus* calcineurin signaling and the cell wall integrity (CWI) pathway**
To assess the mechanism of action of BRI, a collection of 58 protein kinase inhibitors (PKI, at a concentration of 20 μM; Supplementary Table S2) was screened for *A. fumigatus* growth and corresponding metabolic activity alone or together with 20 μM BRI (Supplementary Table S2). Two PKIs, a p21-Activated Kinase Inhibitor FRAX486 and a STK25 inhibitor PP121, both members of the sterile 20 superfamily of kinases, are identified as potentiating the BRI activity against *A. fumigatus* (Fig. 3a, b). The p21 activated kinases (PAKs) belong to the family of Ste20-related kinases and these kinases have been shown to be involved in signaling through mitogen activated protein kinase (MAPK) pathways[60]. The closest STK25 homologs are cAMP-mediated signaling proteins Sok1p in *Saccharomyces cerevisiae* whose overexpression suppresses the growth defect of mutants lacking protein kinase A activity[61]. We also screened a library of 25 *A. fumigatus* null mutants for phosphatase catalytic subunits[62] for sensitivity to BRI 20 μM. We identified a single phosphatase mutant, Δ*calA* (*calA* encodes the calcineurin catalytic subunit), as more sensitive to BRI 20 μM (Supplementary Table S3).

Considering the importance of calcineurin in the signaling response to *A. fumigatus* osmotic stress and the cell wall integrity pathway[63–66] and the fact that PAKs have been shown to be involved in signaling through MAPK pathways[60], we tested the BRI growth inhibition of four *A. fumigatus* null mutants for MAPK Δ*sakA*, Δ*mpkC*, Δ*mpkB* and Δ*mpkA* which regulate osmotic stress and CWI pathways responses[66,67]. We identified only Δ*mpkA* as being more sensitive to BRI 20 μM (Supplementary Table S3). Both mutants Δ*calA* and Δ*mpkA* have severe growth defects (Fig. 3c) and further validation by using Alamar blue metabolic activity assay showed that Δ*calA* strain had similar metabolic activity in the presence or absence of BRI (Fig. 3d). In contrast, the Δ*mpkA* mutant had a significantly decreased metabolic activity when grown in the presence of BRI (Fig. 3d). Hsp90 chaperone is important for the regulation of several fungal signaling pathways either by physical interaction or regulation of their expression[68]. *A. fumigatus* calcineurin and CWI kinases are possible Hsp90 client proteins[69–71]. We investigate the influence of Hsp90 on BRI activity by determining the relationship between the Hsp90 inhibitor geldanamycin (GEL) and BRI in a checkerboard assay for which GEL concentrations of 0 to 25 μg/ml and BRI concentrations of 0 to 80 μM were used. The FIC index for these two drugs was 0.64 indicating additive effect for GEL when added to BRI against *A. fumigatus* (Supplementary Fig. S1).

Cyclosporine (CsA) is a specific inhibitor of calcineurin and there is synergy, in inhibition of *A. fumigatus* metabolic activity, between increasing concentrations of CsA and BRI 20 μM (Fig. 3e). *A. fumigatus* calcineurin regulates the activity of the CrzA transcription factor with the phosphorylated form accumulating in the cell cytosol, and in response to several stimuli, including CAS exposure, calcineurin dephosphorylates CrzA leading to its re-localization to the nucleus[63–65]. When a functional CrzA:GFP strain is not exposed to any drug, or exposed to BRI (5 μM), or to a sub-inhibitory concentration of CAS (0.07 μg/ml), 0, 0 and 28.3 %, respectively, of the germlings have CrzA:GFP in the nuclei, while 57.4 % are in the nuclei when this strain is exposed to a combination of BRI + CAS (Fig. 3f). *A. fumigatus* protein kinase C (PKC) is important for the activation of the CWI pathway[72] and PKC inhibitors such as chelerythrine and calphostin C also synergize with BRI 20 μM (Fig. 3g, h).

To further validate the involvement of BRI in the CWI pathway mediated by MpkA, and CalA, we comparatively evaluated several phenotypic parameters related to this pathway in the wild-type and mutant strains. Caspofungin shows a fungistatic activity not only against the *A. fumigatus* wild-type but also against Δ*mpkA*, and Δ*calA*

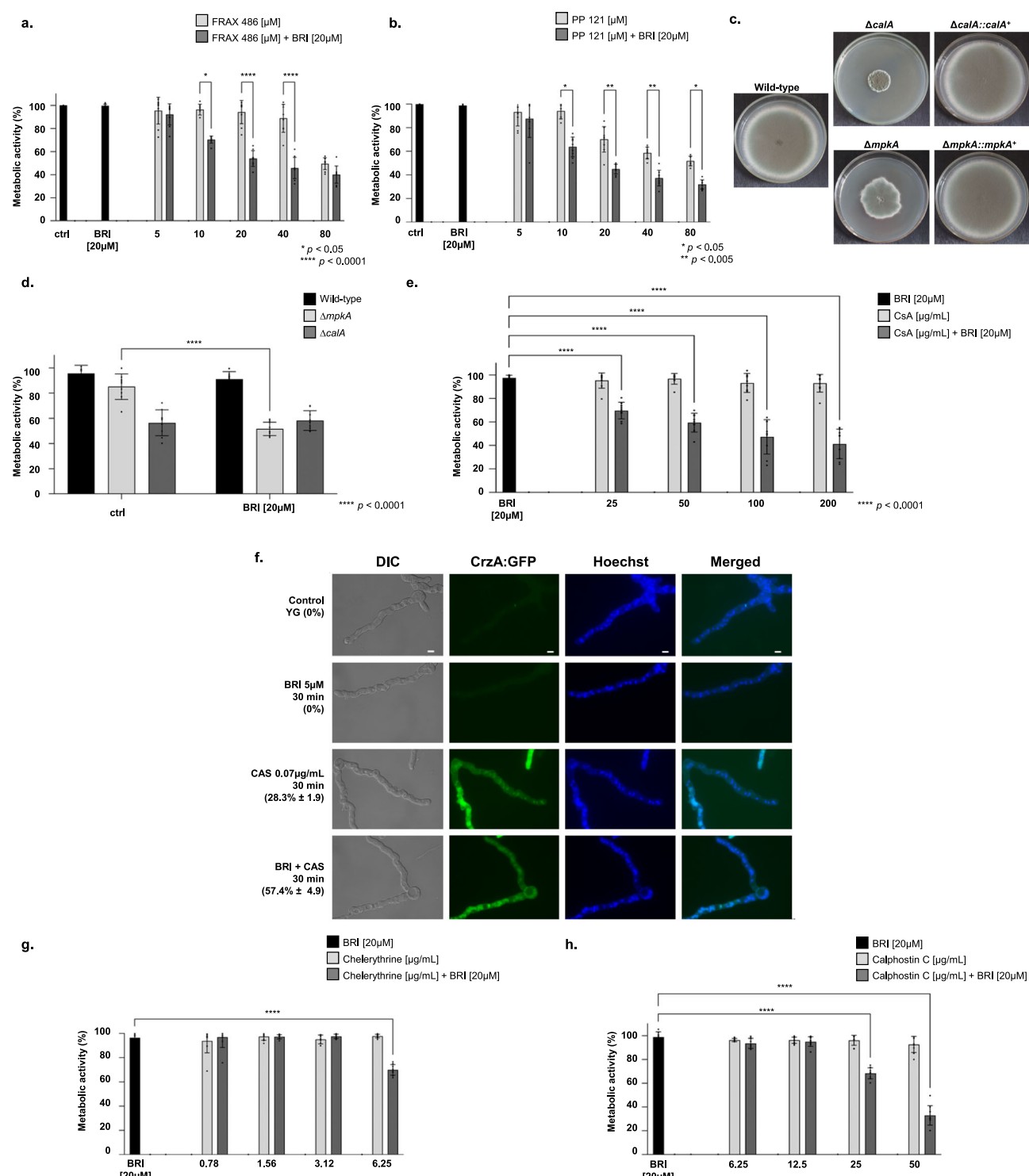

**Fig. 3 | Calcineurin and the MAPK MpkA are important for the BRI + CAS synergism. a** Metabolic activity expressed by Alamar blue of *A. fumigatus* grown for 48 h in the absence or presence of BRI [20 μM], FRAX486 [5–80 μM] or BRI [20 μM] + FRAX486 [5 to 80 μM]. **b** Metabolic activity expressed by Alamar blue of *A. fumigatus* grown for 48 h in the absence or presence of BRI [20 μM], PP121 [5 to 80 μM] or BRI [20 μM] + PP121 [5 to 80 μM]. **c** Growth of the wild-type, Δ*calA*, Δ*calA::calA⁺*, Δ*mpkA*, and Δ*mpkA::mpkA⁺* on MM for 5 days at 37 °C. **d** Metabolic activity expressed by Alamar blue of *A. fumigatus* wild-type, Δ*calA*, and Δ*mpkA* grown for 48 h in the absence or presence of BRI [20 μM]. **e** Metabolic activity expressed by Alamar blue of *A. fumigatus* grown for 48 h in the absence or presence of BRI [20 μM], cyclosporin (CsA) [25 to 200 μg/mL] or BRI [20 μM] + CsA [25 to 200 μg/mL]. **f** CrzA:GFP translocation to the nucleus when germlings were exposed

to BRI 5 μM, CAS [0.07 μg/mL] or BRI [5 μM +CAS 0.07 μg/mL]. Conidia were germinated in YG medium and grown for 13 h and then exposed or not to BRI, CAS, and BRI + CAS for 30 min. Bars, 5 μm. **g** Metabolic activity expressed by Alamar blue of *A. fumigatus* grown for 48 h in the absence or presence of BRI [20 μM], chelerythrine [0.78 to 6.25 μg/mL] or BRI [20 μM] + chelerythrine [0.78 to 6.25 μg/mL]. **h** Metabolic activity expressed by Alamar blue of *A. fumigatus* grown for 48 h in the absence or presence of BRI [20 μM], Calphostin C [6.25–50 μg/mL] or BRI [20 μM] + Calphostin C [6.25–50 μg/mL]. All experiments were independently repeated at least three times. The results with Alamar blue were performed with $n = 3$ and were statistically analysed by 2way ANOVA, Tukey´s post-test (*$p < 0.05$, **$p < 0.005$, ***$p < 0.0005$, and ****$p < 0.0001$).

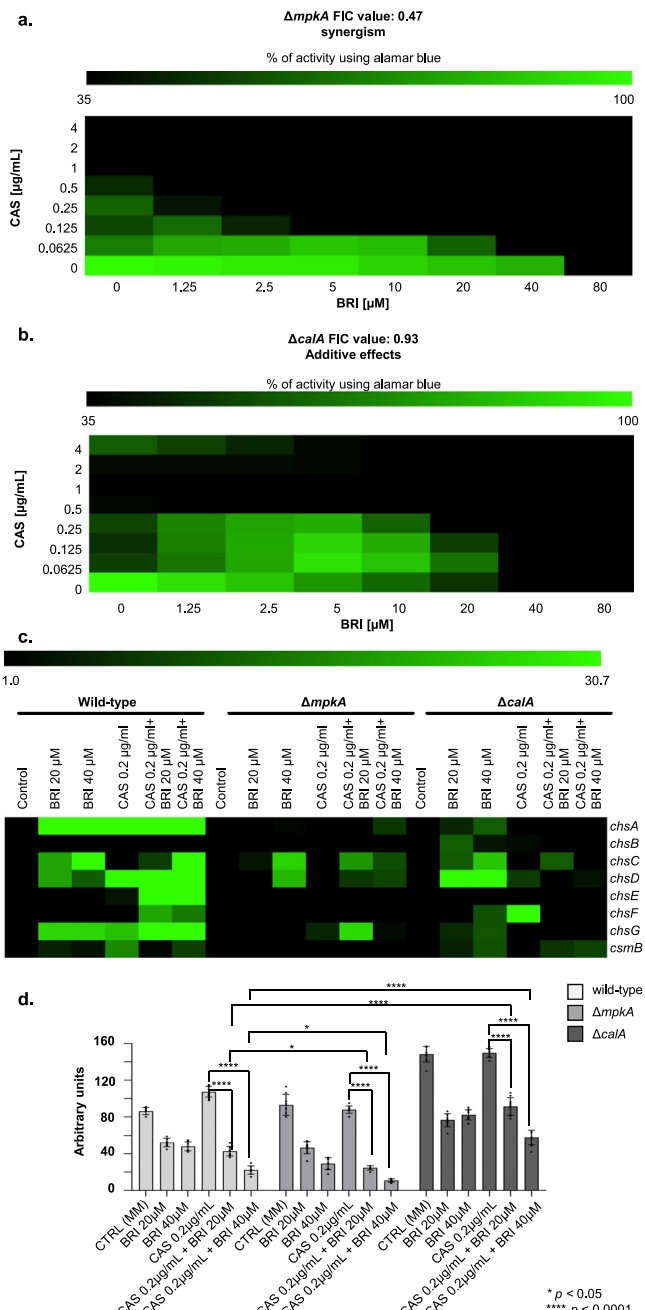

**Fig. 4 | BRI acts in *A. fumigatus* by affecting the cell wall integrity pathway.** **a, b** The Fractional Inhibitory Concentration (FIC) index for BRI + CAS in the Δ*mpkA* and Δ*calA* mutant strains. The results are the average of three independent experiments. **c** Chitin synthase mRNA accumulation in *A. fumigatus* wild-type, Δ*mpkA* and Δ*calA* strains. *A. fumigatus* strains were grown for 24 h at 37 °C and exposed or not to CAS [0.2 μg/mL CAS] combined or not with different concentration of BRI for 1 h. The results are the average of three independent experiments analyzed in technical duplicates each. **d** Exposure of chitin and as measured by Calcofluor White (CFW). The results graphic shows the value of nine technical replicates from three independent experiments ±SD deviation (n = 3; 2way ANOVA, Dunnet´s post-test; *p < 0.05 and ****p < 0.0001).

mutant strains (Supplementary Fig. S2). We compared the relationship between BRI and CAS in a checkerboard assay with the corresponding wild-type and null mutant strains (compare Fig. 2c to Fig. 4a, b). There is an increased susceptibility of Δ*mpkA* and Δ*calA* to lower combinations of BRI + CAS when compared to the wild-type strain (MECs for Δ*calA* and Δ*mpkA* are 0.125 μg/ml and 0.06 μg/ml, respectively,

Fig. 4a, b). The FIC index for these two drugs in the wild-type and the mutant strains is 0.47 and 0.93, indicating synergistic and additive effects for Δ*mpkA* and Δ*calA*, respectively (Fig. 4a, b). Chitin overproduction stabilizes the cell wall at early stages of CAS exposure but this increase by itself is not able to confer resistance to the drug[19]. *A. fumigatus* has 8 chitin synthase genes and the increased expression of some of these genes, a hallmark for the response to CAS[19], was also globally lower in the null mutants than in the wild-type strain when they were exposed to CAS or CAS + BRI (Fig. 4c and Supplementary Fig. S3). Interestingly, not only CAS and CAS + BRI but also BRI alone were able to induce some of these genes in the wild-type strain but not in the mutant strains (Fig. 4c and Supplementary Fig. S3). The germling exposure of chitin, as measured by Calcofluor White (CFW, that recognizes chitin) staining, is dysregulated in the mutant strains: lower in the Δ*mpkA* but higher in Δ*calA* mutants than in the wild-type strain (Fig. 4d).

Taken together, these results strongly indicate that CalA and MpkA are important for BRI activity, and BRI is most likely impacting the *A. fumigatus* CWI pathway.

## BRI potentiates caspofungin activity in *C. neoformans*, *C. albicans*, and *C. auris*

We also investigated if BRI could potentiate CAS activity in other human fungal pathogens, such as *C. neoformans*, *C. albicans*, and *C. auris*. The MICs for BRI in *C. neoformans*, *C. albicans*, and *C. auris* are 2.5 μM, 80 μM and 80 μM, respectively (Table 2). CAS lacks significant activity against *C. neoformans*[73] and only high CAS concentrations, such as CAS 32 μg/mL can completely inhibit *C. neoformans* metabolic activity (as determined by XTT) and CAS 16 μg/mL can decrease survival (colony forming units, CFUs) by about 50 % (Fig. 5a). BRI 0.625 μM (0.25xMIC) potentiates CAS activity (0.25 to 0.5 μg/ml of CAS) resulting in complete inhibition of *C. neoformans* metabolic activity and growth (Fig. 5a). Similarly, complete inhibition of *C. albicans* metabolic activity (XTT) and survival (CFUs) shifted from a concentration of CAS at ~ 0.125 μg/mL in the absence of BRI to a concentration of 0.015 μg/mL of CAS when 20 μM of BRI is added (*i.e.* an 8-fold reduction in the MIC of CAS) (Fig. 5b). BRI also partially suppressed the CAS-resistance of *C. albicans* CAS-resistant clinical isolates (Fig. 5c). BRI MICs for each *C. albicans* CAS-resistant isolate is 20 to 80 μM (Table 2) and the combination of BRI 5 to 20 μM (0.25x MIC) + CAS 0.5 μg/mL decreased CAS-resistance of DPL1006, DPL1007, DPL1009, DPL1010, and DPL1011 7-, 10-, 4-, 10-, and 2-fold, respectively (Fig. 5c). Finally, BRI MIC for *C. auris* is 80 μM (Table 2) and BRI 10 μM + CAS 0.125 μg/mL inhibited about 95 % *C. auris* metabolic activity in clinical isolates 467/2015, 468/2015, 469/2015, 470/2015, and 474/2015 (Fig. 5d). In one of these clinical isolates, 467/2015, BRI 10 μM+CAS 0.5 μg/mL is able to inhibit 100 % metabolic activity and survival, potentiating caspofungin activity by at least 2-fold (Fig. 5d).

Taken together, these results indicate that BRI is able to potentiate CAS activity for different human fungal pathogens, including *C. neoformans*. Interestingly, *C. neoformans* is very sensitive to BRI alone and BRI is fungicidal against this fungus. Thus, BRI is an alternative therapeutic against *C. neoformans* alone or in combination with CAS since it potentiates the latter's activity into a fungicidal drug. BRI + CAS combination has also synergism against *C. auris* dramatically decreasing its viability.

## BRI enhances the activity of POSA against Mucorales fungi

POSA is used for treating patients with lethal mucormycosis as a step-down therapy to liposomal amphotericin B (LAMB) or in lieu of LAMB in patients who are refractory or intolerant to the latter drug[74]. We investigated if BRI could potentiate the activity of POSA against the most common causes of mucormycosis including

**Table 2 | Brilacidin MICs for human pathogenic fungi**

| Species | MIC CAS (μg/ml) | FKS1 mutation | MIC BRI (μM) | Mechanism |
|---|---|---|---|---|
| *Cryptococcus neorformans* H99 | 32.0 | ND | 2.5 | Fungicidal |
| *Candida albicans* SC5314 | 0.25 | ND | 80 | Fungicidal |
| *Candida albicans* DPL1006 CAS[R] | 2.0 | F641L | 20 | Fungicidal |
| *Candida albicans* DPL1007 CAS[R] | 4.0 | F641S | 80 | Fungicidal |
| *Candida albicans* DPL1009 CAS[R] | 4.0 | S645Y | 20 | Fungicidal |
| *Candida albicans* DPL1010 CAS[R] | 2.0 | S645F | 80 | Fungicidal |
| *Candida albicans* DPL1011 CAS[R] | 4.0 | S645F + R1361R/H | 40 | Fungicidal |
| *Candida auris* 467/2015 | 1.0 | ND | 80 | Fungicidal |
| *Candida auris* 468/2015 | 1.0 | ND | 80 | Fungicidal |
| *Candida auris* 469/2015 | 1.0 | ND | 80 | Fungicidal |
| *Candida auris* 470/2015 | 1.0 | ND | 80 | Fungicidal |
| *Candida auris* 471/2015 | 1.0 | ND | 80 | Fungicidal |
| | MIC POSA (μg/ml) | CYP51 mutation | MIC BRI (μM) | Mechanism |
| *Rhizopus oryzae* 99–892 | 2.0 | ND | 32 | Fungicidal |
| *Rhizopus delemar* 99–880 | 2.0 | ND | 32 | Fungicidal |
| *Lichtheimia corymbifera* | 1.0 | ND | 8.0 | Fungicidal |
| *Mucor circineloides* 131 | >16 | ND | 16 | Fungicidal |

Obs: Fungicidal activity was measured by plating the cells or conidia after 48 h incubation with the different drug concentrations on solid minimal medium and calculating the colony forming units divided by the initial inoculum; ND Not determined.

*Rhizopus delemar, R. oryzae, Lichtheimia corymbifera, Mucor circineloides*. The growth of these fungi was measured in a checkerboard 96-well assay for which BRI concentrations of 64-0.12 μg/mL and POSA concentrations of 16–0.25 μg/mL were used. In two independent experiments, the FIC indices for these two drugs ranged between 0.56–0.75 indicating additive effect for BRI when added to POSA against the tested Mucorales fungi (Table 3). Thus, like VOR and BRI against *A. fumigatus*, it appears that BRI can potentiate the activity of POSA against Mucorales fungi in vitro.

### BRI combined with CAS is not toxic to human cells and decreases the *A. fumigatus* fungal burden in a chemotherapeutic murine model

Toxicity assessment of BRI in A549 pulmonary cells (https://bcrj.org.br/, code 0033) was initially performed by incubating the cells either with 40 or 80 μM of BRI with or without increasing CAS concentrations for 48 h, after which cell viability was assessed by XTT assay (Fig. 6a). As a positive control, we used DMSO 10 % that reduced cell viability by 80%. Neither BRI, CAS, or their combinations reduced cell viability when compared to the control (Fig. 6a). We also compared the ability of the combination of BRI + CAS to the standard of care, VOR, in controlling *A. fumigatus* cell growth when infecting A549 pulmonary cells. While VOR, CAS, and BRI monotherapy killed ~60%, 25%, and 0%–10%, respectively, a combination of BRI (20–80 μM) + CAS (100 μg/ml) resulted in 50–85% *A. fumigatus* killing (Fig. 6b).

Finally, we investigated if BRI + CAS could impact *A. fumigatus* virulence in a chemotherapeutic murine model of IPA. Fungal burden in the lungs was ~50% reduced after 3 days post-infection in mice treated either with CAS (1 mg/kg) or BRI (50 mg/kg) when compared with the non-treated mice (Fig. 6c). However, the combination of BRI + CAS significantly reduced the fungal burden by ~95% when compared with the non-treated mice (Fig. 6c). These results strongly indicate that the combination BRI + CAS is able to clear *A. fumigatus* infection in the lungs in a chemotherapeutic murine model of IPA.

Taken together, these data indicate that the combination treatment of BRI + CAS is non-toxic to mammalian cells in vitro and is able to enhance clearance of *A. fumigatus* infection in pulmonary cells in vitro and in vivo when compared to monotherapy alone.

### BRI blocks fungal growth and disease development in a murine model of fungal keratitis

To determine if BRI could be clinically useful in a different disease context, we next employed a treatment model of fungal keratitis. Briefly, ulcerated corneas of C57BL/6 mice were overlayed with *A. fumigatus* swollen conidia in order to establish FK. Starting at 4 h post-inoculation, animals received topical treatment with either (1) 2.5 mM BRI in 12.5% DMSO, (2) 12.5% DMSO (vehicle control) or (3) Tris-buffered saline (TBS). Treatments were staggered by 3 h on any given day, with 3 treatments administered on day 0 and 6 treatments administered on days +1 and +2, with one final treatment on day +3. Uninfected (UI) animals also received the indicated treatments to assess their clinical safety.

Infected (FK) animals treated with TBS or vehicle demonstrated indistinguishable degrees of corneal disease, marked by corneal opacification and surface irregularity at each day post-inoculation (Fig. 7A, C). The presence of clinical disease in those groups corresponded to an increased corneal thickness, relative to UI controls, indicating the presence of corneal edema and inflammation (Fig. 7B, D). By contrast, FK animals treated with BRI were indistinguishable from the UI controls with respect to both pathological readouts at all time points post-inoculation, indicating a complete inhibition of disease development (Fig. 7). Also in contrast to the TBS and vehicle-treated groups, BRI-treated corneas harbored almost no viable fungus (Fig. 7E), suggesting the compound suppressed fungal proliferation within the cornea. However, the high variability of colony forming units in the other treatment groups rendered this observation insignificant by ANOVA ($p = 0.13$). Nevertheless, we conclude that the antifungal activity of BRI in the cornea resulted in a robust and statistically significant reduction in disease pathology. Moreover, uninfected animals receiving TBS, vehicle or BRI treatment were indistinguishable with regards to clinical score, corneal thickness, or histological analysis, suggesting that BRI is not toxic to the ocular surface.

### Discussion

Due to the reduced number of available antifungal agents and the increased emergence of antifungal resistance in the clinical environment, there is an urgent need for novel antifungal drugs. There are few new potential antifungal drugs at different stages of clinical development, such as fosmanogepix (a novel Gwt1 enzyme inhibitor),

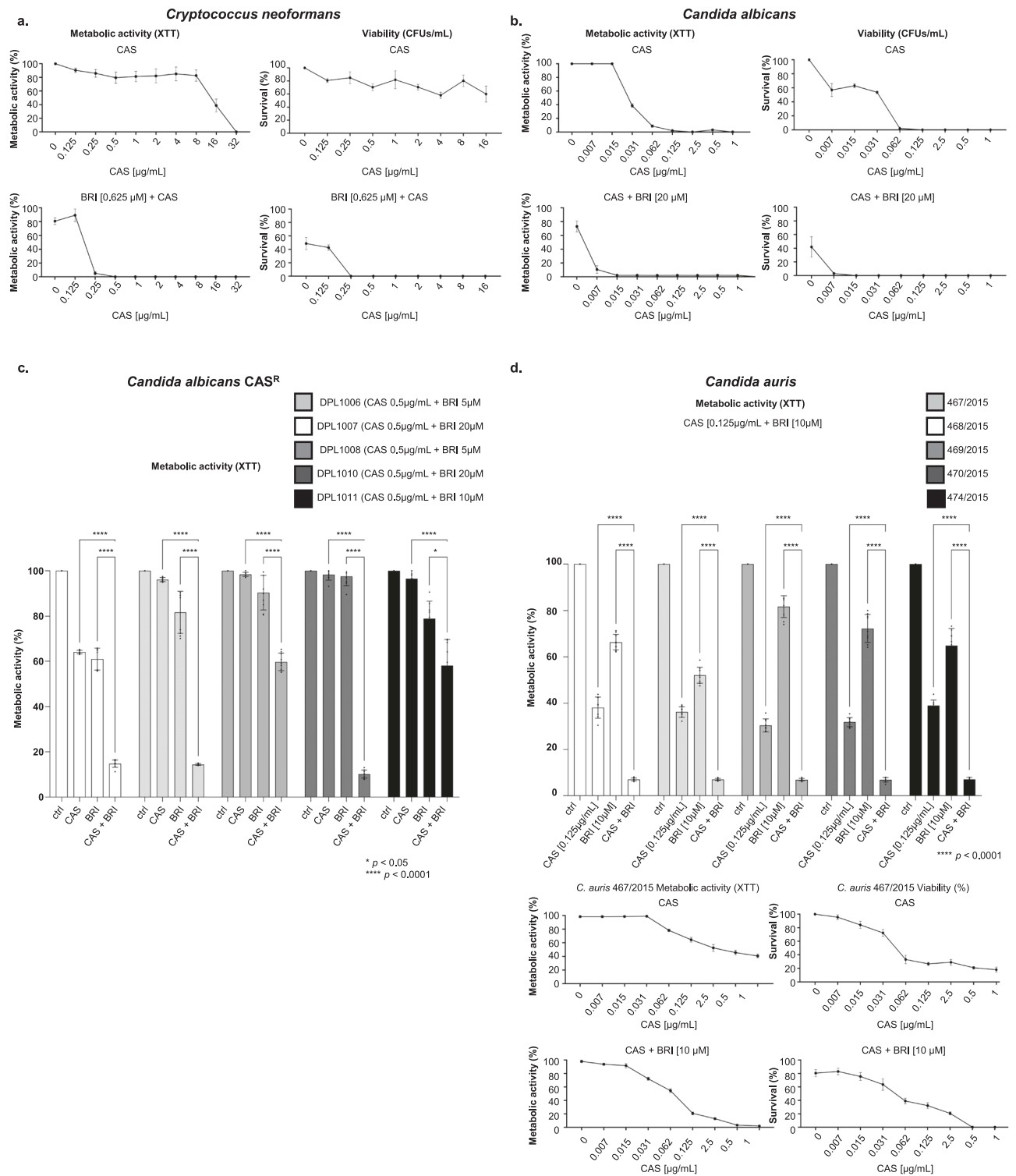

ibrexafungerp (a novel oral glucan synthase inhibitor), olorofim (a novel dihyroorotate dehydrogenase enzyme inhibitor), opelconazole (a novel triazole optimized for inhalation), and rezafungin (an echinocandin with an extended half-life)[28]. An alternative approach to the development of novel antifungal drugs is the repositioning or repurposing of existing drugs. This approach is particularly promising because usually it identifies drugs that can synergize or enhance the activity of current antifungal agents and in many cases overcome developed resistance to a certain class of antifungals. Here, we have screened four repurposing libraries and identified enhancers and synergizers of CAS activity. We concentrated our attention on one of

these synergizers, BRI, that is able to synergize not only with CAS but also VOR. To our knowledge, only two CAS synergizers have been previously identified against *A. fumigatus*, diphenyl diselenide and clofazimine[75,76], and there are no previous reports showing BRI as a potential antifungal agent and synergizer of current antifungal drugs.

Human host defense peptides (HDPs or antimicrobial peptides), one class of which are the human defensins, are part of the innate immune response[77,78]. BRI is a polymer-based, non-peptidic small molecule drug candidate exhibiting antimicrobial and immunomodulatory properties, designed to mimic the structure and activity of

**Fig. 5 | BRI can synergize CAS in other human fungal pathogens. a** Metabolic activity expressed by XTT of *C. neoformans* grown for 48 h in the absence or presence of CAS [0 to 32 μg/mL] or BRI [0.625 μM] + CAS [0−32 μg/mL]. Percentage of survival expressed as colony forming units/ml of *C. neoformans* cells grown for 48 h in the absence or presence of CAS [0 to 16 μg/mL] or BRI [0.625 μM] + CAS [0−16 μg/mL]. Experiments were carried out in three repeats for each sample and three independent experiments ($n = 3$). **b** Metabolic activity expressed by XTT of *C. albicans* grown for 48 h in the absence or presence of CAS [0−1 μg/mL] or BRI [20 μM] + CAS [0 to 1 μg/mL]. Percentage of survival expressed as colony forming units/mL of *C. albicans* cells grown for 48 h in the absence or presence of CAS [0−1 μg/mL] or BRI [20 μM] + CAS [0−1 μg/mL]. Experiments were carried out in three repeats for each sample and three independent experiments ($n = 3$). **c** Metabolic activity expressed by XTT of *C. albicans* CAS-resistant strains grown for 48 h in the absence or presence of CAS [0.5 μg/mL], BRI [5−20 μM] or BRI [5 to

20 μM] + CAS [0.5 μg/mL]. Experiments were carried out in three independent replicates ($n = 3$), with three technical replicates and by analysed by Two-way ANOVA followed by Dunnett´s multiple comparison test (*$p < 0.05$ and ****$p < 0.0001$). The graphic shows the average of the nine repetitions ±SD. **d** Metabolic activity expressed by XTT of *C. auris* grown for 48 h in the absence or presence of CAS [0.125 μg/mL], BRI [10 μM] or BRI [10 μM] + [CAS 0.125 μg/mL]. Metabolic activity expressed by XTT of *C. auris* 467/2015 strain grown for 48 h in the absence or presence of CAS [0 to 1 μg/mL] or BRI [10 μM] + CAS [0 to 1 μg/mL]. Percentage of survival expressed as colony forming units/ml of *C. auris* 467/2015 strain grown for 48 h in the absence or presence of CAS [0 or 1 μg/mL] or BRI [10 μM] + CAS [0 to 1 μg/ml]. Experiments were performed in three independent biological replicates ($n = 3$), with three technical replicates and analysed by Two-way ANOVA followed by Dunnett´s multiple comparison test (*$p < 0.05$ and ****$p < 0.0001$). The graphic shows the average of nine repetitions ±SD.

HDPs[52,79]. BRI has been tested in several clinical trials, such as (i) treatment for acute bacterial skin and skin structure infections (ABSSSI) caused by *Staphylococcus aureus* (Clinical Trials NCT01211470, https://www.clinicaltrials.gov/ct2/show/NCT01211470?term=polymedix[44,80]), (ii) oral mucositis (NCT02324335, https://clinicaltrials.gov/ct2/show/NCT04784897), and more recently (iii) Covid-19/SARS-Cov-2 (https://clinicaltrials.gov/ct2/show/NCT04784897)[81,82]. Like other antimicrobial peptides, BRI can depolarize the *A. fumigatus* cell membrane[83,84]. However, it is not clear from our studies if BRI is localized on the cell surface, or if it can be transported intracellularly and interacts with other cellular proteins. We investigated the mechanism of action for BRI and CAS synergism by a combination of screening PKIs and a collection of *A. fumigatus* null mutants for the catalytic subunits of phosphatases. We observed two PKI compounds, a p21-Activated Kinase Inhibitor FRAX486 and a STK25 inhibitor PP121, both members of the sterile 20 superfamily of kinases, as potentiating the BRI activity against *A. fumigatus*. Since p21 activated kinases (PAKs) are involved in signaling through MAPK pathways[60], we tested if other *A. fumigatus* MAPK are more sensitive to BRI and identified MpkA as being so, the apical MAPK that is regulated by protein kinase C and coordinates the CWI pathway[85]. Since Δ*mpkA* and Δ*calA* have reduced growth when compared to the wild-type strains, these results were confirmed by using chelerythrine and calphostin C (inhibitors of protein kinase C) and cyclosporin (an inhibitor of calcineurin). Complementary to this screening, we also identified increased CrzA:GFP translocation to the nucleus in the presence of BRI + CAS. We demonstrated that the combination of BRI + CAS affects several aspects of the CWI pathway, such as altered increased or decreased expression of chitin synthase genes, and the exposure of chitin to the cell wall, strongly suggesting that BRI is modulating the *A. fumigatus* CWI pathway. Thus, we propose that BRI + CAS has a synergism that is able to affect *A. fumigatus* viability through multiple mechanisms of action, encompassing functional changes/depolarization of the microorganism cell membrane, interference to calcineurin signaling (as for instance the increased CrzA translocation), and misexpression of the cell wall integrity pathway, and necessary compensatory reactions (since CAS is also present and prevents further β-1,3-glucan biosynthesis). Consequently, cell wall assembly and remodeling, which prevents resistance against osmotic forces, is inhibited which then leads to cell lysis. Alterations in the β-1,3-glucan synthase lipid environment could also be important[86]. Recently, a novel off-target

mechanism of action for CAS was proposed in which CAS-induced ROS alters the lipid composition around the β-1,3-glucan synthase, changing its conformation and making it less sensitive to CAS[86]. It is possible that BRI also influences the lipid composition of the cell membrane (e.g., by interacting with ergosterol) making the β-1,3-glucan synthase more accessible to CAS. It is likely that BRI can potentiate VOR activity through the disorganization of the cell membrane which would not allow the proper deposition of ergosterol leading to impaired function of the sterol. An important feature of BRI was its ability to overcome CAS-resistance in both *A. fumigatus* and *C. albicans* but not VOR-resistance in *A. fumigatus*. The molecular basis for CAS-resistance in both species is related to hotspot mutations in the *fks1* gene that encodes β-1,3-glucan synthase[87]. All the above suggested mechanisms could also contribute to the bypass of CAS-resistance by BRI. It remains to be investigated how BRI can bypass CAS-resistance in *A. fumigatus*. Equally important is the finding that BRI can enhance the activity of POSA against Mucorales fungi which cause mucormycosis, a disease with high mortality rates of 50−90%, thereby suggesting a potential enhancement in the therapeutic outcome of this lethal disease.

A remarkable observation of our work is the fact that BRI can potentiate CAS activity not only against *A. fumigatus*, but also *C. albicans*, *C. auris*, and *C. neoformans*. There are very few therapeutic options against the treatment of cryptococcosis[31] and BRI demonstrated activity against *C. neoformans* in low concentrations (MIC = 2.5 μM). *C. neoformans* is intrinsically resistant to CAS and only very high non-physiological CAS concentrations can partially inhibit *C. neoformans* growth[87]. However, *C. neoformans* β-1,3-glucan synthase is very sensitive to CAS[88], which suggests that other mechanisms unrelated to β-1,3-glucan synthase resistance are important for CAS resistance. We have not investigated the molecular basis for BRI potentiation of CAS in *C. neoformans*. Nevertheless, there is a report showing that cyclosporine and FK506 (tacrolimus) can enhance the activity of caspofungin against *C. neoformans*[89], suggesting the possibility that the same mechanism observed for *A. fumigatus* is also acting in *C. neoformans*. Interestingly, amphotericin B which disturbs the cell membrane via self-assembly into an extramembranous sponge that rapidly extracts ergosterol from fungal membranes, synergizes with CAS against *C. neoformans*[90,91]. and Mucorales fungi[92]. This suggests that an analogous mechanism of cell membrane disruption by amphotericin could enhance CAS activity in *C. neoformans*, reinforcing the notion that BRI is impacting the organization of the cell membrane in this species.

Initial studies demonstrated that the combination of BRI + CAS is not toxic to human alveolar basal epithelial cells and can significantly reduce the *A. fumigatus* fungal burden in an immunosuppressed murine model, strongly indicating the therapeutic potential of BRI + CAS against IPA. Additionally, BRI alone also decreases *A. fumigatus* fungal burden and ablates disease development in a murine model of fungal keratitis. Further studies will address possible formulations to deliver BRI alone or BRI and CAS together and to understand the

**Table 3 | FIC indices of two different experiments showing BRI + POSA combination against Mucorales fungi**

| Mucorales Fungi | Experiment 1 | Experiment 2 |
|---|---|---|
| *R. oryzae* 99–892 | 0.75 | 0.75 |
| *R. delemar* 99–880 | 0.56 | 0.75 |
| *L. corymbifera* | 0.62 | 0.72 |
| *M. circineloides* 131 | 0.51 | 0.62 |

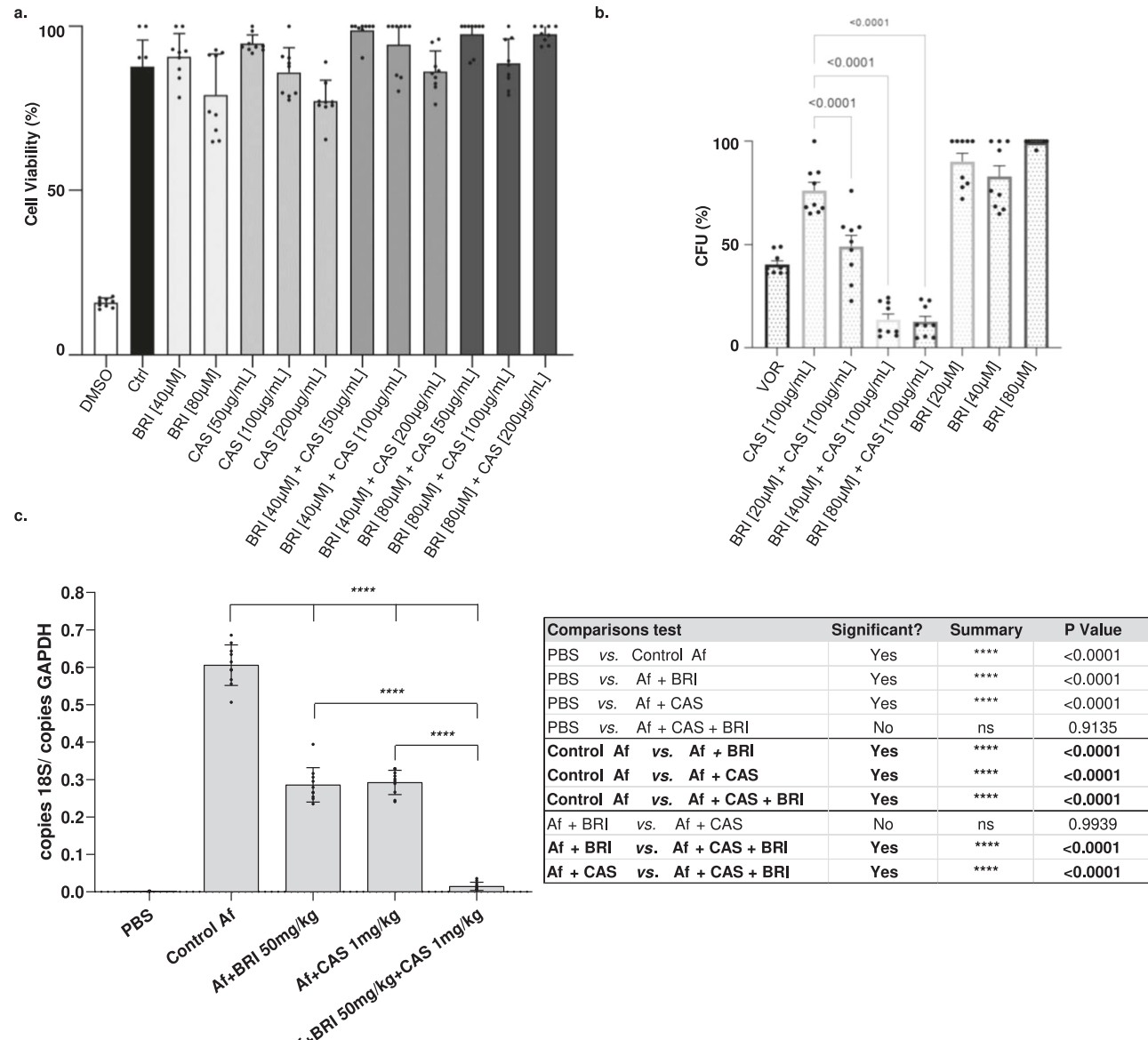

**Fig. 6 | The combination of BRI+CAS is not toxic to human cells and can significantly decrease the *A. fumigatus* fungal burden in a chemotherapeutic murine model. a** A549 lung cells grown in the absence or presence of different concentrations of BRI and CAS. DMSO 10% was used as positive control. Percentage of cell viability is expressed as the absorbance value of experiment well/absorbance value of control well x 100. All the results are the average of three repetitions ±SD. **b** A549 cells were seeded at a density of 106 cells/ml and challenged with *A. fumigatus* conidia at a multiplicity of infection of 1:10 in the absence or presence of different concentrations of BRI and CAS. After 24 h of incubation media was removed and the cell suspension were seeded in Sabouraud Dextrose Agar Media. The numbers of CFUs were determined after 24 h of growth. The CFU percentage for each sample was calculated and the results were plotted using Graphpad Prism (GraphPad Software, Inc., La Jolla, CA, USA). A *p*-value ≤ 0.001 was considered significant. **c** Quantification of the fungal burden in the infected lungs was determined 72 h post-infection. The total genomic DNA was extracted from the lungs, and fungal burden was determined by qPCR based on 18 S rRNA gene of *A. fumigatus* and an intronic region of the mouse GAPDH gene through the ratio between number of copies of the 18 S and number of copies of GAPDH. Data represent the means (±standard deviation) of two biological replicates (*n* = 10, representative of two independent experiments). *P*-values were calculated using One-way ANOVA with Tukey's multiple comparisons test. **$p < 0.0001$.

pharmacokinetics. Our results demonstrate the promise of repurposing studies and the screening of well annotated compounds for potentiating the activity of current antifungal drugs.

# Methods

## Ethical statement

The principles that guide our studies are based on the Declaration of Animal Rights ratified by UNESCO on January 27, 1978 in its 8th and 14th articles. All protocols adopted in this study were approved by the local ethics committee for animal experiments from the University of São Paulo, Campus of Ribeirão Preto (Permit Number: 08.1.1277.53.6; Studies on the interaction of *Aspergillus fumigatus* with animals). Groups of five animals were housed in individually ventilated cages and were cared for in strict accordance with the principles outlined by the Brazilian College of Animal Experimentation (COBEA) and Guiding Principles for Research Involving Animals and Human Beings, American Physiological Society. All efforts were made to minimize suffering. Animals were clinically monitored at least twice daily and humanely sacrificed if moribund (defined by lethargy, dyspnea, hypothermia and weight loss). All stressed animals were sacrificed by cervical dislocation.

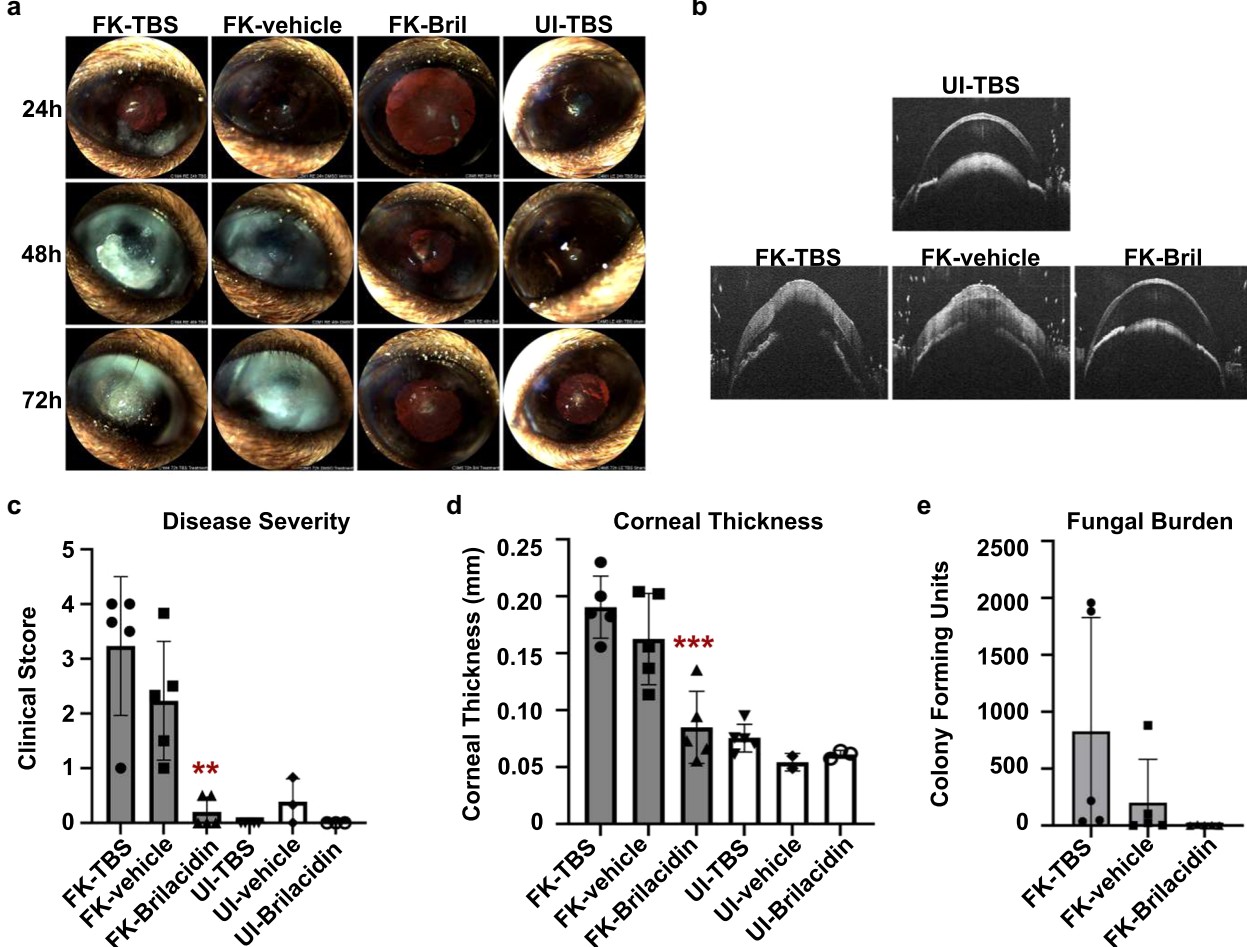

**Fig. 7 | Topical BRI treatment blocks fungal growth and disease development in a murine model of fungal keratitis. a** Representative slit-lamp images of uninfected (UI) or *A. fumigatus* infected (FK, fungal keratitis) murine corneas. Animals were treated multiple times daily with drops containing Tris-buffered saline (TBS), vehicle (TBS + 12.5% DMSO), or brilacidin (2.5 mM Bril + vehicle) as indicated. **b** Representative optical coherence tomography (OCT) scans of the indicated groups at 72 h p.i. **c** Disease scoring of all corneas at 72 h p.i. based on the slit-lamp images. **d** Thickness measurements of all corneas at 72 h p.i. derived from OCT analysis. **e** Colony forming unit measurements from corneal homogenates taken from all corneas at 72 h p.i. The data in all panels are from a single experiment; $n = 5$ corneas for all FK groups; $n = 6$ for UI-TBS; $n = 3$ for UI-vehicle and UI-Bril groups. For **c**–**e**, all groups were compared by one-way ANOVA, **($p < .01$), ***($p < .001$). The ability of Bril treatment to eliminate viable fungus and block disease development was similarly observed in a preliminary experiment using the same number of animals but with a different treatment dosage and frequency.

## Strains, media and cultivation methods

The *A. fumigatus. Candida* spp.*, C. neoformans* strains and Mucorales fungi used in this work are listed in Supplementary Table S4. *Aspergillus* strains were grown in minimal medium (MM; 1% [wt/vol] glucose, 50 mL of 20x salt solution, trace elements, 2% [wt/vol], pH 6.5. For solid minimal medium 2% agar was added) at 37 °C. Solutions of trace elements and salt solution are described[93]. For the animal studies, *A. fumigatus* strain (wild-type, Δ*ku80 pyrG*+) was grown on MM. Fresh conidia were harvested in PBS and filtered through a Miracloth (Calbiochem). Conidial suspensions were spun for 5 min at 3000x g, washed with PBS, counted using a hemocytometer and resuspended at a concentration of $5.0 \times 10^7$ conidia/mL. *Candida* spp. and *C. neoformans* strains were grown and maintained on YPD (1% yeast extract, 2% peptone and 2% glucose). For Mucorales fungi, all organisms were grown on potato dextrose agar (PDA; BD Diagnostic, New Jersey) plates by sprinkling silica beads that have been absorbed with spore suspension of each organism. The plates were incubated at 37 °C for 4–7 days. Spores were collected in endotoxin-free Dulbecco phosphate buffered saline (PBS) containing 0.01% or 0.2% Tween 80 for Mucorales or *Aspergillus*, respectively. Collected spores were washed

with PBS, and counted with a hemocytometer to prepare the final inocula.

## Library drug screenings

The Pandemic Response box (400 compounds) and COVID box (160 compounds) (both available at https://www.mmv.org), the National Institutes of Health (NIH) clinical collection (727 compounds) (available at https://pubchem.ncbi.nlm.nih.gov/source/ NIH%20Clinical%20Collection) and the SGC's epigenetic chemical probe and SGC's pharma donated chemical probe libraries (115 compounds, see https://www.thesgc.org/chemical-probes) totaling 1,402 compounds, were screened in the current study. The primary screening was performed against the *A. fumigatus* wild-type strain by using the chemical libraries diluted in dimethyl sulfoxide (DMSO). The ability of each compound in combination (or not) with caspofungin (CAS) in blocking the fungal growth was visually determined. Briefly, each well of a flat-bottom polystyrene microplate was filled with 198 μL of liquid MM containing $1 \times 10^4$ conidia/mL of *A. fumigatus* (wild-type strain). Subsequently 20 μM of each chemical compound was added in combination (or not) with 0.2 μg/

mL of CAS to each well. This concentration represents the MEC for CAS against *A. fumigatus*. Plates were statically incubated for 48 h at 37 °C. Wells containing only medium, CAS [0.2 µg/mL] or DMSO were used as controls. Compounds presenting over 80% of visual fungal growth inhibition (in combination or not with CAS) were selected for further studies. All experiments were done in triplicate.

## Alamar blue assays

The inhibition of the metabolic activity of *A. fumigatus* triggered by the drugs selected in the first screening was assessed by using Alamar blue (Life Technologies) according to ref. 94. The experiment was done by inoculation of 100 µL of liquid MM containing $2.5 \times 10^3$ conidia/mL of the *A. fumigatus* wild-type strain supplemented or not with CAS [0.2 µg/mL] plus increasing concentration of each selected drug [0.6–20 µM] and 10% Alamar blue in 96-well plates. As positive controls, the drugs were replaced by the same volume of the medium. As the negative control, wells were filled with 90 µL of liquid MM plus 10 µL of Alamar Blue. Plates were incubated for 48 h at 37 °C without shaking and results were read spectrophotometrically by fluorescence (570 nm excitation– 590 nm emission) in a microplate reader (SpectraMax® Paradigm® Multi-Mode Microplate Reader; Molecular Devices). Enhancers were defined as compounds that alone inhibited over 30% of *A. fumigatus* metabolic acitivity but in combination with CAS inhibited even more, while synergizers were defined as compounds which alone inhibited <30% of the fungal metabolic activity but in combination with CAS inhibited >30%.

A protein kinase inhibitors (PKI) library was also screened in combination with BRI. In total, 58 PKI were analyzed. Briefly, 100 µL of liquid MM containing $2.5 \times 10^3$ conidia/mL of the *A. fumigatus* wild-type strain plus 10% alamar blue was inoculated with increasing concentration of PKI [5–80 µM] in the presence (or not) of BRI [20 µM] and incubated 48 h at 37 °C without shaking. To analyze the sensitivity of the null mutants Δ*calA*, Δ*mpkA* and their complementing strains to BRI, 100 µL of liquid MM containing $2.5 \times 10^3$ conidia/mL of each strain was inoculated in the presence (or not) of BRI [20 µM] plus 10% alamar blue and incubated 48 h at 37 °C without shaking. To check if cyclosporine (CsA), chelerythrine and/or calphostin C synergize with BRI, variable concentrations of each one of these drugs was analyzed in the presence of BRI [20 µM]. A total 100 µL of liquid MM containing $2.5 \times 10^3$ conidia/mL of the *A. fumigatus* wild-type strain plus 10% alamar blue was inoculated with increasing concentrations of CsA [25–200 µg/mL], chelerythrine [0.78–6.25 µg/mL] and calphostin C [6.25–50 µg/mL] in the presence (or not) of BRI [20 µM]. Plates were incubated for 48 h at 37 °C without shaking. All experiments containing alamar blue were, after 48 h incubation, read spectrophotometrically by fluorescence and analyzed as previously described by[94]. Experiments were repeated at least three times.

## Minimal inhibitory concentration (MIC)

The BRI drug used for MIC assays was kindly supplied by Dr. William DeGrado and solubilized in DMSO. The minimal inhibitory concentration (MIC) of BRI for *A. fumigatus* or Mucorales fungi was determined based on the M38-A2 protocol of the Clinical and Laboratory Standards Institute[95] and for yeasts using M27-A3 method[96]. In brief, the MIC assay was performed in 96-well flat-bottom polystyrene microplate where 200 µL of a suspension ($1 \times 10^4$ conidia/mL) prepared in liquid MM was dispensed in each well and supplemented with increasing concentration of BRI (ranging from 0 to 160 µM). Plates were incubated at 37 °C without shaking for 48 h and the inhibition of growth was evaluated. The MIC was defined as the lowest drug concentration that visually attained 100% of fungal growth inhibition compared with the control well. Wells containing only MM and DMSO were used as a control. Similar protocol was used for yeast organisms, except that we used RPMI-1640, $1 \times 10^3$ cells/mL/well and incubated the plates for 48 h (*Candida* spp.) or 72 h (*C. neoformans*).

## FIC index analysis

To determine synergy, additive, indifference, or antagonism between BRI and CAS, BRI and VOR, and geldanamycin and BRI (against *A. fumigatus*) and BRI and POSA (against Mucorales fungi), we used the FIC index method[97]. Briefly, for all of the wells of the microtitration plates that corresponded to an MIC, the sum of the FICs (ΣFIC) was calculated for each well with the equation ΣFIC = FICA + FICB = (CA(comb)/MICA(alone)) + (CB(comb)/MICB(alone)), where MICA(alone) and MICB(alone) are the MICs of drugs A and B when acting alone, and CA(comb) and CB(comb) are the concentrations of the drugs A and B at the iso-effective combinations. An FIC index of <0.5 indicates synergism, >0.5–1 indicates additive effects, >1 to <2 indifference, and ≥2 is considered to be antagonism[98].

## Combination of BRI and CAS against yeasts

For measuring the effect of the combination BRI + CAS against yeast fungal pathogens, two methods were used: (i) metabolic activity by XTT-assay as described by ref. 99 and (ii) colony forming units (CFUs). For the first method, *C. neoformans*, *C. albicans*, and *C. auris* $10^4$ cells were inoculated in RPMI-1640 supplemented with CAS 0–32 µg/ml (for *C. neoformans*) and 0 to 1 µg/ml (for *Candida* spp.) or the same concentrations of CAS combined with BRI 0.625 µM (for *C. neoformans*), BRI 20 µM (for *C. albicans*), and BRI 10 µM (for *C. auris*). After 48 h of incubation, the viable cells were revealed using XTT-assay as described by[99]. XTT-assays were also used for *C. albicans* caspofungin resistant strains but with CAS 0.5 µg/ml combined with BRI 5, 10 or 20 µM. The same experimental design was used for the CFUs determination, except that after 48 h the cells present in the wells were plated on YPD (yeast extract 10 g, peptone 20 g, dextrose 20 g, agar 20 g, water 1000 ml) and the plates were incubated at 30 °C for 24–48 h for determining the survival percentage. The results are the average of three repetitions and are expressed as average ± standard deviation.

## Conidial viability exposed to BRI, VOR and CAS

The viability of *A. fumigatus* conidia exposed to CAS + BRI or VOR + BRI was assessed by plating the cells after being treated. Initially, a suspension containing $1 \times 10^4$ conidia/ml of *A. fumigatus* cells was prepared in liquid MM and 200 µL of this suspension was dispensed in each well of a 96-well polystyrene microplate supplemented with CAS [0.2 or 0.50 µg/mL] + BRI [20 µM] or VOR [0.125 or 0.25 µg/mL] + BRI [20 µM]. After 48 h incubation at 37 °C, a total of 100 conidia was plated in solid complete medium (YAG) [2% (w/v) glucose, 0.5% (w/v) yeast extract, trace elements] or minimal medium [1% (w/v) glucose, nitrate salts, trace elements, pH 6.5] and let to grow at 37 °C for 36 h. The number of viable colonies was determined by counting the number of colony-forming units (CFUs) and expressed in comparison with the negative control (no germinated and untreated conidia), which gives 100% survival. Results are expressed as means and standard deviations (SD) from three independent experiments.

## Biofilm assay

To test the susceptibility of pre-formed *A. fumigatus* biofilms to VOR, CAS and to the combination of CAS + BRI and VOR + BRI, a suspension containing $10^6$ conidia per mL of the wild-type strain (Δ*ku80 pyrG*⁺) was prepared in liquid MM and 100 µL of it was inoculated in each well of a 96-well plate. After 24 h of incubation at 37 °C, 50 µL of fresh MM containing CAS, VOR or the combination of VOR and CAS with BRI was added to the biofilm to reach the final concentration as indicated and incubated for a further 12 h at 37 °C. Wells containing untreated conidia were used as a positive control. After, the metabolic activity of the cells was evaluated by adding 50 µL of an aqueous XTT solution (1 mg/mL of XTT and 125 µM of menadione) to each well. The plate was incubated for an additional 1 h at 37 °C, centrifuged (2000 rpm, 5 min) and 100 µL of the supernatant was transferred to a flat-bottomed 96-well plate. The absorbance was measured at 450 nm on a plate reader

(Synergy HTX Multi-Mode Reader- BioTek Instruments). The XTT assays were performed in six replicates.

## Staining for chitin

This procedure was performed as described by refs. [62], [100], [101]. Briefly, *A. fumigatus* conidia ($2 \times 10^4$) were grown in 200 μL of MM for 16 h at 37 °C. The germlings were exposed or not to different concentrations of BRI, CAS or BRI + CAS for 4 h and the culture medium was removed and the germlings were UV-irradiated (600,000 μJ). For chitin staining, 200 μL of a PBS solution with 10 μg/mL of CFW were added to UV-irradiated germlings, incubated for 5 min at room temperature, washed three times with PBS and fluorescence was read at 380 nm excitation and 450 nm emission. All the experiments were performed with eight repetitions and fluorescence was read in a microtiter plate reader Synergy/HTX multi-mode reader (Biotek).

## Phosphatase and kinase null mutant screening

An *A. fumigatus* phosphatase deletion library encompassing 25 null mutants for phosphatase catalytic subunits[90] was screened for sensitivity to the combination of CAS + BRI. *A. fumigatus* null mutants for MAPK (Δ*sakA*, Δ*mpkC*, Δ*sakA*; Δ*mpkC*, Δ*mpkB* and Δ*mpkA)* were also screened. The assay was performed in a 96-well flat-bottom polystyrene microplate. In each well a total of 200 μL of liquid MM plus conidia from the different mutants ($1 \times 10^4$ conidia/mL) was incubated in the presence of BRI [20 μM]. Plates were incubated at 37 °C without shaking for 48 h and the inhibition of growth was visually evaluated. Wells containing only MM and DMSO were used as a control.

## Membrane potential determination

The effect of the CAS [0.125 μg/mL], BRI [1 μM] or the combination CAS + BRI ([0.125 μg/mL] and [1 μM], respectively) on the cell membrane potential was assessed by using the bis-(1,3-dibutylbarbituric acid) trimethine oxonol−DiBAC4(3) reagent (Invitrogen, Carlsbad, CA, USA) according to[54]. with modifications. *A. fumigatus* conidia were inoculated on coverslips in 5 mL of liquid MM and cultivated for 16 h at 30 °C. Further, coverslips containing adherent germlings were left untreated or treated with CAS, BRI or CAS + BRI plus 3 μg/ml DIBAC$_4$(3) and incubated for 30 min at 30 °C in the dark. After, the germlings were washed with sterile PBS (140 mM NaCl, 2 mM KCl, 10 mM NaHPO4, 1.8 mM KH2 PO4, pH 7.4). The fluorescence was analyzed with excitation wavelength of 450nm−490 nm, and emission wavelength of 525 nm−550 nm on the Observer Z1 fluorescence microscope (Carl Zeiss) using the 100x with Differential interference contrast (DIC) images. Fluorescent images were captured with an AxioCam camera (Carl Zeiss, Inc.) and processed using the AxioVision software (version 4.8). In each experiment, at least 50 germlings were counted and the experiment repeated at least three times.

## RNA extraction, cDNA synthesis and RTqPCR

All experiments were carried out in biological triplicates and conidia ($10^7$) were inoculated in liquid MM and *A. fumigatus* strains were grown for 16 h at 37 °C and treated or not with different concentrations of BRI, CAS, or BRI + CAS for 1 h. For total RNA isolation, mycelia were ground in liquid nitrogen and total RNA was extracted using TRIzol (Invitrogen), treated with RQ1 RNase-free DNase I (Promega), and purified using the RNAeasy kit (Qiagen) according to the manufacturer's instructions. RNA was quantified using a NanoDrop and Qubit fluorometer, and analyzed using an Agilent 2100 Bioanalyzer system to assess the integrity of the RNA. All RNA had an RNA integrity number (RIN) between 8.0 and 10 (Thermo Scientific) according to the manufacturer's protocol.

For RT-qPCR, the RNA was reverse transcribed to cDNA using the ImProm-II reverse transcription system (Promega) according to manufacturer's instructions, and the synthesized cDNA was used for real-time analysis using the SYBR green PCR master mix kit (Applied Biosystems) in the ABI 7500 Fast real-time PCR system (Applied Biosystems, Foster City, CA, USA). Primer sequences are listed in Supplementary Table S5.

## Cytotoxicity assay

Cytotoxicity assays in A549 human lung cancer cells were performed using XTT assay as indicated in the manufacturers' instructions. Cells ($2 \times 10^5$ cells/well) were seeded in 96-well tissue plates and incubated in Dulbecco's Modified Eagle Medium (DMEM) culture medium. After 24 h of incubation, the cells were treated with BRI (40 and 80 μM/well), CAS (50, 100 and 200 μg/well) or in different CAS + BRI combinations. After 48 h incubation, the cell viability was assessed by using the XTT kit (Roche Applied Science) according to the manufacturer's instructions. Formazan formation was quantified spectrophotometrically at 450 nm (reference wavelength 620 nm) using a microplate reader. The experiment was made in three replicates. Viability was calculated using the background-corrected absorbance as follows: Cell viability (%) = absorbance value of experiment well/absorbance value of control well x 100.

## Killing assay

The type II pneumocyte cell line A549 was cultured using DMEM (ThermoFischer Scientific, Paisley, UK) supplemented with 10% fetal bovine serum (FBS) and 1% penicillin−streptomycin (Sigma-Aldrich, Gillingham, UK) and seeded at a density of $10^6$ cells/ml in 24-well plates (Corning). The cells were treated with BRI (20, 40 and 80 μM/well), CAS (100 μg/well) or in different combinations between them and challenged with *A. fumigatus* conidia at a multiplicity of infection of 1:10. After 24 h of incubation in 5% $CO_2$ the culture media was removed, and 2 ml of sterile water was added to the wells. A P1000 tip was then used to scrape away the cell monolayer and the cell suspension was collected. This suspension was then diluted 1:1000 and 100 μl was plated on Sabouraud Dextrose Agar Media before the plates were incubated a 37 °C overnight. The numbers of CFUs were determined after 24 h of growth. A volume of 50 μl of the inoculum adjusted to $10^3$/ml was also plated on SAB agar to correct CFU counts. The CFU percentage for each sample was calculated and the results were plotted using Graphpad Prism (GraphPad Software, Inc., La Jolla, CA, USA). A *p*-value ≤0.001 was considered significant.

## Fungal burden

Inbred female mice (BALB/c strain; body weight, 20–22 g obtained from the facility of the campus of the Universidade de São Paulo, Ribeirão Preto, Brazil) were housed in vented cages containing five animals. Cages were well ventilated, softly lit and subjected to 12:12 hs light-dark cycle. The relative humidity was kept at 40 to 60%. Mouse rooms and cages were kept at a temperature of 22 °C. Mice were immunosuppressed with cyclophosphamide (150 mg/kg of body weight), which was administered intraperitoneally on days −4, −1 and 2 prior to and post infection (infection day is "day 0"). Hydrocortisonacetate (200 mg/kg body weight) was injected subcutaneously on day -3. Mice (five mice per group) were anesthetized by halothane inhalation and infected by intranasal instillation of 20 μL of $1.0 \times 10^6$ conidia of *A. fumigatus* Δ*ku80 pyrG*+(wild-type) (the viability of the administered inoculum was determined by incubating a serial dilution of the conidia on MM medium, at 37 °C). As a negative control, a group of five mice received PBS only. On the same day of infection (day 0), mice received concomitantly the first dose of treatment with BRI (50 mg per kg of body weight) and/or CAS (1 mg per kg of body weight), administered intraperitoneally. The second dose of drugs was administered 24 h after infection. Animals were sacrificed 72 h postinfection, and the lungs were harvested and immediately frozen in liquid nitrogen. Samples were lyophilized and homogenized by vortexing with glass beads for 5 min, and DNA was extracted via the phenol/chloroform method.

DNA quantity and quality were assessed using a NanoDrop 2000 spectrophotometer (Thermo Scientific). Quantitative real-time PCRs were performed using 400 ng of total DNA from each sample, and primers to amplify the 18 S rRNA region of *A. fumigatus* and an intronic region of mouse GAPDH (glyceraldehyde-3-phosphate dehydrogenase). Six-point standard curves were calculated using serial dilutions of gDNA from *A. fumigatus* strain and the uninfected mouse lung. Fungal and mouse DNA quantities were obtained from the threshold cycle (Ct) values from an appropriate standard curve.

### Murine model of fungal keratitis

**Fungal inoculum.** On the day of inoculation, *A. fumigatus* (Af293) conidia were inoculated into 25 mL of YPD broth at a density of $5 \times 10^6$/ mL, and incubated for 4 h at 35 °C, 200 RPM. The swollen (but not polarized) conidia were collected by centrifugation, washed with PBS (2x), and resuspended in PBS to an optical density of 0.8 (360 nm). This inoculum was stored at room temperature until corneal inoculation (-1 h later).

**Establishment of FK.** On the day preceding corneal inoculation (day -1), 6–8 week old male C57BL/6 mice (Jackson Laboratories, body weight, 20–22 g) were injected i.p. with 100 mg/kg methylprednisolone. On day 0, animals were anesthetized by i.p. injection of 100 mg/ kg ketamine and 6.6 mg/kg xylazine. While anesthetized, the central corneal epithelium of the right eye was ulcerated to a diameter of -1 mm with an Algerbrush II and 5 uL of the fungal inoculum (see above) was pipetted onto the corneal surface. 20 min following instillation, the inoculum was removed with a Kim wipe. The contralateral eye of each animal remained uninfected in accordance with the Association for Research in Vision and Ophthalmology guidelines for the use of animals in vision research. At the time of inoculation, animals were administered 1 mg/kg Buprenorphine SR by subcutaneous injection for analgesia. Cages were well ventilated, softly lit and subjected to a 12:12 hs light-dark cycle. The relative humidity was kept at 40 to 60%. Mouse rooms and cages were kept at a temperature of 22 °C.

**Topical BRI treatment of FK animals.** Fungal or sham-inoculated mice were divided into groups to receive treatment with the following: (1) 2.5 mM BRI in 12.5% DMSO, (2) 12.5% DMSO (vehicle controls), or (3) Tris-buffered saline (TBS). For each treatment, 5 uL drops were applied to the corneal surface transiently for 5 min to mice under isoflurane anesthesia. The drug is applied as a drop to the ocular surface and remains there with the animal under isoflurane anesthesia for 5 min. When the animal regains consciousness, the blink reflex and movement will naturally cause the drop to drain away. On day 0, animals received 3 treatments, starting at 4 h p.i. and repeated every 3 h. On day +1 and day +2, animals received 6 treatmetns, each 3 h apart. On day +3, animals received a single treatment in the morning before final imaging, euthanasia, and ocular resection for fungal burden analysis.

**Slit-lamp biomicroscopy and disease scoring of corneas.** At each day p.i., mice were anesthetized with isoflurane and photographed using a Micron IV biomicroscope (Phoenix Technology Group, USA). Images were deindentified and assigned an overall disease score (range 0–4) by two blinded reviewers based on server criteria, including (1) area of opacification, (2) density of opacification, and (3) surface irregulating. Average disease scores for each cornea were analyzed by one-way ANOVA.

**Optical coherence tomography and corneal thickness measurement.** Each day p.i., animals were anesthetized with isoflurane and corneas were imaged using a Bioptigen spectral-domain optical coherence tomography system (Leica Microsystems, USA). Reference arm calibration was completed by the manufacturer and set to 885. For each cornea, a $4 \times 4$ mm image was scanned with a 12-mm telecentric lens. Images were analyzed using the InVivoVue Diver software (Bioptigen). Corneal scans were digitally overlaid with a 11 x 11 spiderplot, and the distance between the epithelium and endothelium was measured at 11 distinct points near the central cornea, the average of which was was taken as the corneal thickness (in millimeters). Groups were compared using a one-way ANOVA.

**Fungal burden from corneas.** At 72 h p.i., corneas were incubated in 1 mL buffer containing 2 mg/mL collagenase 1 (Sigma) for 1 h at 37 °C. Dilutions of the homogenates were plated onto inhibitory mold agar, incubated overnight at 35 °C, and colonies were enumerated. Colony counts were compared across groups using a one-way ANOVA.

### Statistical analysis

Grouped column plots with standard deviation error bars were used for representations of data. For comparisons with data from wild-type or control conditions, we performed one-tailed, paired *t*-tests or one-way analysis of variance (ANOVA). All statistical analyses and graphics building were performed by using GraphPad Prism 5.00 (GraphPad Software).

### Reporting summary

Further information on research design is available in the Nature Portfolio Reporting Summary linked to this article.

## Data availability

All the data are available as supplementary tables and figures. Drug compounds were obtained from the Pandemic Response box (400 compounds) and COVID box (160 compounds) (both available at https://www.mmv.org), the National Institutes of Health (NIH) clinical collection (727 compounds) (available at https://pubchem.ncbi.nlm. nih.gov/source/ NIH%20Clinical%20Collection) and the SGC's epigenetic chemical probe and SGC's pharma donated chemical probe libraries (115 compounds, see https://www.thesgc.org/chemical-probes) totaling 1402 compounds. Source data are provided with this paper.

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

## Acknowledgements

We thank the Fundação de Amparo à Pesquisa do Estado de São Paulo (FAPESP) grants numbers 2016/07870-9 and 2021/04977-5 (G.H.G.) and the Conselho Nacional de Desenvolvimento Científico e Tecnológico (CNPq) grant numbers 301058/2019-9 and 404735/2018-5 (G.H.G.), both from Brazil, and the National Institutes of Health/National Institute of Allergy and Infectious Diseases grants R01AI153356, and R01AI063503 to ASI from the USA. A special thanks to CAPES for providing the grant for the postdoctoral position for Pedro F. N. Souza. The epigenetic and donated chemical probe libraries were supplied by the Structural Genomics Consortium under an Open Science Trust Agreement: http://www.thesgc.org/click-trust. We thank Drs. Katrien Lagrou, David Perlin, Arnaldo L. Colombo, and Iran Malavazi for providing *Aspergillus fumigatus*, *Candida albicans* and *Candida auris* clinical isolates, and primers for the chitin synthase experiments. We also thank Jane Harness and Kyle Weston for critical reading of the manuscript.

## Author contributions

T.f.d.R., P.A.d.C., R.W.B., C.F.P., P.F.N.S., A.S.I., J.D.L., E.M.A., S.Al., and K.K.F. performed most of the experiments. S.Ac., M.A.H., D.H.D., H.J., and W.f.d. provided materials and synthesized reagents for the work. G.H.G. analyzed the data, wrote the manuscript and coordinated all the work. All the authors read and edited the manuscript.

## Competing interests

W.F.d. is a member of the scientific advisory board of Innovation Pharmaceuticals, a company that is conducting clinical trials on brilacidin. Other authors declare no competing interests.
