## [Peer review file · Nature Communications]

REVIEWER COMMENTS

Reviewer #1 (Remarks to the Author):

The paper entitled 'A host defense peptide mimetic, brilacidin, potentiates caspofungin antifungal activity against human pathogenic fungi' screened a collected compounds library and identified BRI as a synergizer with caspofungin (CAS) for treating fungal infections. Furthermore, the authors hypothesized the acting mechanism of BRI, revealing that BRI blocked the complementation pathway of CAS (CaIA-CrzA-MpkA). Finally, the authors tested the toxicity and effectiveness of BRI in cells and animal model levels.

Despite BRI displayed well synergistic antifungal activity with CAS in vitro and in vivo, partial experiments are not well designed, and some conclusions are decisive and farfetched. The overall draft should be improved in terms of logic, consistency and brevity, and is not qualified for publishing in NC.

Comments:

Abstract

Data not solid enough to support "disrupting cell membrane potential and impairing cell wall integrity pathway".

Introduction

1. As this manuscript focuses on the combination of BRI and CAS, it would be better to justify why finding drugs showing synergic effect with CAS is better?
2. Organizing the action mechanism as a separate paragraph rather than mixing with the last paragraph, which should summarize key findings of this study (e.g., the synergic effect of BRI, the toxicity and effectiveness in murine model).
3. Calcineurin and the MpkA-mediated CWI pathways should be summarized.

Results

1. The first section describes outcomes of compound screening. Please use the title directly reflects key findings rather than "screening of the covid box...". Also this section includes a lot of detailed information that should be put as supplementary information. Not all hyperlinks of compounds or compound library are available, please check.
2. Please provide references for the definition of enhancers and synergizers, please provide more information why BRI is worth focusing in this study. *A. fumigatus* strains treated by CAS displayed short but highly branched hyphae growth, when combine with BRI, what's the phenotype? Could the authors provide images to show fungicidal or fungistatic effect?
3. How does the acting hypothesis of BRI lands on CWI? Any evidence for CAS fungistatic activity through CaIA/CrzA and MpkA? RT-PCR and WB should be applied to validate the changes of the mentioned

pathways, as well as cell membrane ergosterol content and cell wall polysaccharides content analysis. Definition for fungicidal and fungistatic is farfetched.

4. Was the mechanism of BRI potentiates azoles the same as CAS? Why CAS+BRI works for both CAS- and VOR-resistant strains, but VOR+BRI does not work for VOR-resistant strains?

5. In chemotherapeutic murine model, why CEA17 strain is chosen? This is a pyrG auxotroph strain that could not grow without uridine/uracil supplementation, therefore is not suitable for virulence test. In contrast, in murine model of fungal keratitis (FK), strain Af293 was applied, could the authors give explanations? More surprisingly, in FK model 12.5% DMSO was used as the control and for treatment, this is too high to believe! At this concentration, DMSO itself is highly toxic to fungal strains!

Discussion

1. More advanced information needs to be discussed. For instance,

a. Is this the first time to identify compounds showing synergic effect with CAS?

b. Is this the first time that BRI is used in antifungal study?

2. BRI alone can work in the FK model, but it cannot inhibit the growth of *A. fumigatus* on its own. Why “paradoxical” outcomes observed from different experiments?

Figures

1. Fig. 1 the MMV1634399 had a better effect than others. Why didn't choose MMV1634399 for subsequent research?

2. Fig. 2B, the FIC index for BRI+CAS was 0.39, but was 0.34 in the text..

3. Fig.5A misses a positive control showing full toxicity against the cell line. Fig. 5C p values for which two groups comparison is not clear, please indicate significance in text

Tables

1. Table 1: strain CYP-15-184 to CYP-15-147 could be merged into one line with footnote, thus to make the table more concise and informative. The MEC value of CAS was 0.25 µg/mL (except for 3 CAS-resistant strains), but in Fig. 1A, the MEC value of CAS was 0.2 µg/mL, so which is correct?

2. Table 2: How fungicidal mechanism was judged? Footnote for ND is not shown

3. Table 3: Why shown experiment 1 and experiment 2?

Materials and Methods

“An FIC index of < 0.5 indicates synergism, > 0.5-1 indicates additive effects, > 1 to < 2 indifference, and ≥ 2 is considered to be antagonism”. But in the “BRI enhances the activity of POSA against Mucorales fungi” section, the FIC value of *Rhizopus delemar* was 0.56, in table 3 it means additive effects. However, in experiment 2, for the same fungus, the FIC value changed to 1.12, it means indifference. Please explain why the results of the two experiments were inconsistent.

Reviewer #2 (Remarks to the Author):

This manuscript addresses a critical need in fungal research: the development of novel therapies to treat fungal diseases. Here the authors screen small chemical collections for compounds that increase the activity of Caspofungin, a clinically relevant treatment for Aspergillosis. They identified several enhancers and potentiators of caspofungin but focus on characterization of one (Brilicidin) and its interaction with caspofungin. Overall, this work nicely demonstrates an interaction between the host defense mimetic brilacidin (BRI) and caspofungin and shows efficacy in murine models of invasive pulmonary aspergillosis and fungal keratitis. Though the results of the paper are compelling and generally well executed there are several instances where the text does not reflect the data presented. There are a few other minor comments to be addressed however upon addressing these I feel this paper is of significant value to the field and should be published.

The following comments should be addressed:

1. On page 9 “Taken together, these results indicate, converting CAS from a fungistatic to fungicidal drug...” This statement has not been adequately addressed experimentally. Figure 2A attempts to address the mode of killing, however this experiment is missing controls (a comparison of CAS/VORI alone to the combination treatments) and does not actually address whether these combination treatments are fungicidal against hyphal cells which are the cells that the drug is acting on during infection. This is particularly important because caspofungin works by lysing hyphal tips; although this is lethal to each tip compartment, the overall activity of the drug is not cidal. Thus, using conidia to address cidity versus static activity is not relevant to the activity of the drug.
2. In table 1, there are symbols used for “little growth” and “partial inhibition”; it is not clear what little growth means and how it is different from partial inhibition.
3. Figure 3C does not add much to the figure and could easily be added to supplementary.
4. There are no methods stated as to how ‘fungicidal’ activity was assessed for the zygomycetes species tested in Table 2.
5. Page 10 “ These results strongly suggest that CrzA is important for synergistic activity of BRI+CAS” has not really been experimentally addressed. Only the sensitivity to BRI was tested for calA and mpkA – is the interaction between CAS and BRI still synergistic against these strains? It is clear that the

combination treatment activates the CWIP through the nuclear localization of CrzA and therefore the CWIP probably plays a role in response to these treatments.

6. Page 13 “Thus, unlike VOR and BRI against *A. fumigatus*, it appears that the BRI can potentiate the activity of POSA against Mucorales fungi in vitro” This conclusion is perplexing because the FICI results for both the Mucorales and *A. fumigatus* are the same. Both interactions yielded additive FICI values between 0.5 and 1. What is the basis for the conclusion that the combination treatment is different between these fungi?

7. The materials and methods (and Table 1) state “Wild type *A. fumigatus*” was used but then CEA17 is listed – my assumption is this is a mistake and should say CEA10? CEA17 is a uridine and uracil auxotroph; there is no mention of supplementation in the paper and the use of this strain in animals would not be appropriate since it does not grow in mouse lungs.

8. The *calA* results are contradictory. In the discussion the authors state “Complementary to this screening, we also identified calcineurin to be more sensitive to BRI” however Figure 3D shows the opposite result where the same concentration of BRI has no effect on the *calA* mutant.

9. Similar to the above comment, some of the discussion is contradictory to the results in the paper - “it remains to be investigated how BRI can potentiate VOR” though none of the results of the paper suggest there is any interaction with voriconazole including the FIC checkerboard assays and the inability of BRI to overcome VOR resistance.

10. It is unclear if the proper statistics have been applied throughout the paper. None of the figures describe what tests were used and what the comparison for each test is except for Figure 6. The Statistical Analysis portion of the Materials and Methods only mentions a t-test which is not appropriate for all the analyses in the paper. Particularly the analysis of Figure 5 which includes sets of data with multiple comparisons and animal data for which a t-test may or may not be appropriate (proper tests should be used to determine whether parametric vs non-parametric is appropriate). In figure 3D the important comparison would be whether the effect of BRI treatment on *mpkA* or *calA* is different from that of WT; it might be better to present these data as the change in growth (/metabolism) with the addition of BRI to directly compare the effects between strains, given that this mutant strain already has a defect compared to WT. The actual difference in the treatment effect does not seem to be that large (~10%).

Reviewer #3 (Remarks to the Author):

Fernanda Dos Reis et al analyzed the synergy of brilacidin, a mimetic of a host defense peptide and caspofungin. The work is well done and the data are sound. However, the work lacks novelty

1. The topic is not new and the synergy between antifungals and host derived peptides has been shown previously. Indeed, looking for synergy between antifungal molecules has been recommended for many years and lead to many publications.
2. The increased antifungal efficacy between the inhibition of the CWI pathway (mainly due to specific gene deletions) and echinocandins is well known. Moreover, in these early publications, the role of phosphatases regulating the Hog an CWI pathways has been shown.
3. The direct activity of brilacidin +/-caspofungin against beta 1,3 glucan synthase activity has not been biochemically investigated.
4. The finding that this peptide is synergistically active against all human pathogenic fungi in combination with caspofungin and azoles questions the specificity of the mode of action of this mimetic. Is this synergistic effect due more to the activity of this peptide on membrane depolymerization rather than on the CWI pathway?

Answers to the Reviewers

Reviewer #1:

The paper entitled 'A host defense peptide mimetic, brilacidin, potentiates caspofungin antifungal activity against human pathogenic fungi' screened a collected compounds library and identified BRI as a synergizer with caspofungin (CAS) for treating fungal infections. Furthermore, the authors hypothesized the acting mechanism of BRI, revealing that BRI blocked the complementation pathway of CAS (CaIA-CrzA-MpkA). Finally, the authors tested the toxicity and effectiveness of BRI in cells and animal model levels. Despite BRI displayed well synergistic antifungal activity with CAS in vitro and in vivo, partial experiments are not well designed, and some conclusions are decisive and farfetched. The overall draft should be improved in terms of logic, consistency and brevity, and is not qualified for publishing in NC.

Answer: We thank the reviewer for the comments and suggestions to improve the manuscript. We have revised the manuscript to improve the logic, consistency and brevity as suggested.

Comments:

Abstract

Data not solid enough to support "disrupting cell membrane potential and impairing cell wall integrity pathway".

Answer: The sentence was replaced by: "...affecting cell membrane potential and cell wall integrity pathway."

Introduction

1. As this manuscript focuses on the combination of BRI and CAS, it would be better to justify why finding drugs showing synergic effect with CAS is better?

Answer: We thank the reviewer for the suggestion. We added a sentence (Page 5, lines 109 to 111): "For an example, an efficient approach to control aspergillosis could be the identification of compounds that can potentiate CAS to becoming fungicidal".

2. Organizing the action mechanism as a separate paragraph rather than mixing with the last paragraph, which should summarize key findings of this study (e.g., the synergic effect of BRI, the toxicity and effectiveness in murine model).

Answer: All the introduction was reorganized to reflect the recommendations of the reviewer.

3. Calcineurin and the MpkA-mediated CWI pathways should be summarized.

Answer: We thank the reviewer for the suggestion. We have added a few sentences explaining the involvement of the pathways upon caspofungin exposure (Pages 4 and 5, lines 91 to 98 and 99, respectively): “Specifically, caspofungin affects the composition and organization of the *A. fumigatus* cell wall with hyphae hyperbranching, lysis of hyphal apical compartments, resulting in loss of cell wall β -1,3-glucan. In the presence of CAS, the cell wall integrity (CWI) pathway is activated and the CWI mitogen-activated protein kinase MpkA and the phosphatase calcineurin turn on the transcription factors RlmA and CrzA, which translocate to the nucleus and regulate the activation of several stress responses and cell wall modifications, including chitin synthase gene expression resulting in overproduction of chitin^{19,20}.”

Results

1. The first section describes outcomes of compound screening. Please use the title directly reflects key findings rather than “screening of the covid box...”. Also this section includes a lot of detailed information that should be put as supplementary information. Not all hyperlinks of compounds or compound library are available, please check.

Answer: We thank the reviewer for the suggestion. We change the name of the section to (Page 6, lines 141): **“Identification of a host defense peptide mimetic, BRI, as a CAS synergizer.”**

If the reviewer does not mind, we would prefer to keep all this detailed data in the section. All the four hyperlinks described in this section for compounds or compound libraries are functional on my internet browser. Maybe the popup must be activated in the reviewer’s internet browser.

2. Please provide references for the definition of enhancers and synergizers, please provide more information why BRI is worth focusing in this study. *A. fumigatus* strains treated by CAS displayed short but highly branched hyphae growth, when combine with BRI, what’s the phenotype? Could the authors provide images to show fungicidal or fungistatic effect?

Answer: We thank the reviewer for the suggestions. We have provided more references for the definition of enhancers and synergizers (Page 7, lines 165 to 166): “(for definitions and applications of synergizers and enhancers, see ³⁸⁻⁴¹).”

We also have added why BRI is worth focusing in this study (Page 8, lines 209 to 219): “Although other compounds, like MMV1634399, showed the highest potentiation of CAS activity, we decided to concentrate our further analysis on the host defense peptide mimetic brilacidin (BRI) since this drug candidate has undergone several clinical trials showing therapeutic benefit with acceptable safety profiles. BRI is a small molecule, new chemical entity, created to mimic the amphiphilic structure of host defense proteins (HDPs), having one surface with positively charged groups (cationic) and the opposite surface consisting of hydrophobic groups⁴⁹⁻⁵². With this general synthetic form, there is no need for an agent to be of the size or composition of naturally-occurring proteins to effectively

function as a HDP, as the ability to act as a HDP is retained by the much smaller synthetic amphiphilic molecule.”

We have provided microscopic images of the effects of brilacidin combined with caspofungin in Figure 2B.

3. How does the acting hypothesis of BRI lands on CWI? Any evidence for CAS fungistatic activity through CalA/CrzA and MpkA? RT-PCR and WB should be applied to validate the changes of the mentioned pathways, as well as cell membrane ergosterol content and cell wall polysaccharides content analysis. Definition for fungicidal and fungistatic is farfetched.

Answer: We thank the reviewer for the excellent suggestions. We have performed additional experiments: (i) checkerboard assays for the $\Delta mpkA$ and $\Delta calA$ mutant strains, (ii) RTqPCRs for chitin synthases, and (iii) measurement of the exposure of chitin in the cell walls. These experiments are described in Figure 4 and in the text:

(Page 12, lines 339-361): “To further validate the involvement of BRI in the CWI pathway mediated by MpkA, and CalA, we comparatively evaluated several phenotypic parameters related to this pathway in the wild-type and mutant strains. Caspofungin shows a fungistatic activity not only against the *A. fumigatus* wild-type but also against $\Delta mpkA$, and $\Delta calA$ mutant strains (Supplementary Figure S2). We compared the relationship between BRI and CAS in a checkerboard assay with the corresponding wild-type and null mutant strains (compare Figure 2C to Figures 4A and 4B). There is an increased susceptibility of $\Delta mpkA$ and $\Delta calA$ to lower combinations of BRI+CAS when compared to the wild-type strain (Figures 4A and 4B). The FIC index for these two drugs in the wild-type and the mutant strains is 0.47 and 0.93, indicating synergistic and additive effects for $\Delta mpkA$ and $\Delta calA$, respectively (Figures 4A and 4B). Chitin overproduction stabilizes the cell wall at early stages of CAS exposure but this increase by itself is not able to confer resistance to the drug¹⁹. *A. fumigatus* has 8 chitin synthase genes and the increased expression of some of these genes, a hallmark for the response to CAS¹⁹, was also globally lower in the null mutants than in the wild-type strain when they were exposed to CAS or CAS+BRI (Figure 4C and Supplementary Figure S3). Interestingly, not only CAS and CAS+BRI but also BRI alone were able to induce some of these genes in the wild-type strain but not in the mutant strains (Figure 4C and Supplementary Figure S3). The germling exposure of chitin, as measured by Calcofluor White (CFW, that recognizes chitin) staining, is dysregulated in the mutant strains: lower in the $\Delta mpkA$ but higher in $\Delta calA$ mutants than in the wild-type strain (Figure 4D).”

Caspofungin displays a fungistatic activity against the wild-type and the mutants and this is shown now in the text and in the Supplementary Figure S2 (Page 12, lines 339 to 344): ““To further validate the involvement of BRI in the CWI pathway mediated by MpkA, and CalA, we comparatively evaluated several phenotypic parameters related to this pathway in the wild-type and mutant strains. Caspofungin shows a fungistatic activity not only against the *A. fumigatus* wild-type but also against $\Delta mpkA$, and $\Delta calA$ mutant strains (Supplementary Figure S2).”.

If the reviewer does not mind, we would prefer to focus on the determination of mechanism of action of the synergism between brilacidin and caspofungin, and not perform the determination of ergosterol content.

4. Was the mechanism of BRI potentiates azoles the same as CAS? Why CAS+BRI works for both CAS- and VOR-resistant strains, but VOR+BRI does not work for VOR-resistant strains?

Answer: Unfortunately, we cannot provide an explanation for that. Additional studies beyond the scope of the current manuscript are necessary to understand this phenomenon.

5. In chemotherapeutic murine model, why CEA17 strain is chosen? This is a pyrG auxotroph strain that could not grow without uridine/uracil supplementation, therefore is not suitable for virulence test. In contrast, in murine model of fungal keratitis (FK), strain Af293 was applied, could the authors give explanations? More surprisingly, in FK model 12.5% DMSO was used as the control and for treatment, this is too high to believe! At this concentration, DMSO itself is highly toxic to fungal strains!

Answer: We apologize about the mistake. The name of the strain is wrong. We have used a prototrophic strain $\Delta ku80$ pyrG⁺ and this was modified across the manuscript.

Although, 12.5 % displays some toxicity to *A. fumigatus in vitro*, this concentration was not able to inhibit *A. fumigatus* growth *in vivo* since the control of the fungal keratitis experiment (vehicle) use the same DMSO concentration and the fungus was able to grow and infect the mice cornea. Brilacidin can inhibit *A. fumigatus* growth at a MIC higher than 80 μ M. The fact that brilacidin can inhibit *A. fumigatus* growth in the fungal keratitis experiment is because the brilacidin concentration used in these experiments is supra-MIC concentration. We have to treat the eyes at a high concentration because the drug is only exposed to the ocular surface transiently. This is why the DMSO concentration is so high, *i.e.*, we are using close to the stock concentration. But, most topical antifungals (natamycin and voriconazole) are at a high concentration. For example, 1% voriconazole drops is equal to 28 mM, which much more above the MIC. This is now explained in the Materials and Methods section (Page 28, lines 873 to 877): "For each treatment, 5 μ L drops were applied to the corneal surface transiently for 5 minutes to mice under isoflurane anesthesia. The drug is applied as a drop to the ocular surface and remains there with the animal under isoflurane anesthesia for five minutes. When the animal regains consciousness, the blink reflex and movement will naturally cause the drop to drain away."

Discussion

1. More advanced information needs to be discussed. For instance,
a. Is this the first time to identify compounds showing synergic effect with CAS?

b. Is this the first time that BRI is used in antifungal study?

Answer: We thank the reviewer for the comment. We have added this information to the Discussion section (Page 16, lines 483 to 486): “To our knowledge, only two CAS synergizers have been previously identified against *A. fumigatus*, diphenyl diselenide and clofazimine^{76,77}, and this is the first time that BRI is shown as a potential antifungal agent and synergizer of current antifungal drugs.”

2. BRI alone can work in the FK model, but it cannot inhibit the growth of *A. fumigatus* on its own. Why “paradoxical” outcomes observed from different experiments?

Answer: Brilacidin can inhibit *A. fumigatus* growth at a MIC higher than 80 μ M, (WT, Table 1 and Supplementary Table S1). The fact that brilacidin can inhibit *A. fumigatus* growth in the fungal keratitis experiment is because the brilacidin concentration used in these experiments is supra-MIC. We have to treat the eyes at a high concentration because the drug is only exposed to the ocular surface transiently. Most topical antifungals (natamycin and voriconazole) are at a high concentration. For example, 1% voriconazole drops is equal to 28 mM, way above the in vitro MIC.

Figures

1. Fig. 1 the MMV1634399 had a better effect than others. Why didn't choose MMV1634399 for subsequent research?

Answer: We thank the reviewer for the observation. We added a sentence explaining the reason why we choose brilacidin instead of MMV1634399 (Page 8, lines 209 to 219): “Although other compounds, like MMV1634399, showed the highest potentiation of CAS activity, we decided to concentrate our further analysis on the host defense peptide mimetic brilacidin (BRI) since this drug candidate has undergone several clinical trials showing therapeutic benefit with acceptable safety profiles. BRI is a small molecule, new chemical entity, created to mimic the amphiphilic structure of host defense proteins (HDPs), having one surface with positively charged groups (cationic) and the opposite surface consisting of hydrophobic groups⁴⁹⁻⁵². With this general synthetic form, there is no need for an agent to be of the size or composition of naturally-occurring proteins to effectively function as a HDP, as the ability to act as a HDP is retained by the much smaller synthetic amphiphilic molecule.”

2. Fig. 2B, the FIC index for BRI+CAS was 0.39, but was 0.34 in the text..

Answer: This was now modified in the text.

3. Fig.5A misses a positive control showing full toxicity against the cell line. Fig. 5C p values for which two groups comparison is not clear, please indicate significance in text Tables

Answer: Answer: We have added a positive control with 10 % DMSO to now Figure 6A. This is now stated in the text (Page 14, lines 416 to 417): “As a positive control, we used DMSO 10 % that reduced cell viability by 80% (Figure 6A).”

We have added a text table next to the now Figure 6C showing the different comparisons and significant *p*-values.

1. Table 1: strain CYP-15-184 to CYP-15-147 could be merged into one line with footnote, thus to make the table more concise and informative. The MEC value of CAS was 0.25 µg/mL (except for 3 CAS-resistant strains), but in Fig. 1A, the MEC value of CAS was 0.2 µg/mL, so which is correct?

Answer: If acceptable to the reviewer and editor, we would prefer to keep the Table 1 without merging the lines. We believe this will help the reader to identify which individual strain and its corresponding phenotype. The MEC value of CAS is 0.25 µg/mL but our screening was performed with 0.20 µg/mL, these values are correct in Figures 1 and 2.

2. Table 2: How fungicidal mechanism was judged? Footnote for ND is not shown

Answer: We have added a footnote to the Table 2, addressing it: “Obs: *Fungicidal activity was measured by plating the cells or conidia after 48 hours incubation with the different drug concentrations on solid minimal medium and calculating the colony forming units divided by the initial inoculum; ND=Not determined”

3. Table 3: Why shown experiment 1 and experiment 2?

Answer: Experiment 1 and experiment 2 reflect the results of two independent experiments. In these kind of experiments, we cannot express the results as an average and standard deviation. This is now explained in the title of the Table 3: “Table 3. FIC indices of two different experiments showing BRI + POSA combination against Mucorales fungi”.

Materials and Methods

“An FIC index of < 0.5 indicates synergism, > 0.5-1 indicates additive effects, > 1 to < 2 indifference, and ≥ 2 is considered to be antagonism”. But in the “BRI enhances the activity of POSA against Mucorales fungi” section, the FIC value of *Rhizopus delemar* was 0.56, in table 3 it means additive effects. However, in experiment 2, for the same fungus, the FIC value changed to 1.12, it means indifference. Please explain why the results of the two experiments were inconsistent.

Answer: We have repeated this experiment twice now (we have now four different experiment values), and they showed FICs with values of 0.56, 1.12, 0.75, and 0.75. Since three of these experiments gave values > 0.5-1, we consider this interaction as additive and have included the values of 0.56 and 0.75 in Table 3 and in the text.

Reviewer #2:

This manuscript addresses a critical need in fungal research: the development of novel therapies to treat fungal diseases. Here the authors screen small chemical collections for compounds that increase the activity of Caspofungin, a clinically relevant treatment for Aspergillosis. They identified several enhancers and potentiators of caspofungin but focus on characterization of one (Brilacidin) and its interaction with caspofungin. Overall, this work nicely demonstrates an interaction between the host defense mimetic brilacidin (BRI) and caspofungin and shows efficacy in murine models of invasive pulmonary aspergillosis and fungal keratitis. Though the results of the paper are compelling and generally well executed there are several instances where the text does not reflect the data presented. There are a few other minor comments to be addressed however upon addressing these I feel this paper is of significant value to the field and should be published.

Answer: We thank the reviewer for the comments and for the recognition of the importance of our work.

The following comments should be addressed:

1. On page 9 “Taken together, these results indicate, converting CAS from a fungistatic to fungicidal drug...” This statement has not been adequately addressed experimentally. Figure 2A attempts to address the mode of killing, however this experiment is missing controls (a comparison of CAS/VORI alone to the combination treatments) and does not actually address whether these combination treatments are fungicidal against hyphal cells which are the cells that the drug is acting on during infection. This is particularly important because caspofungin works by lysing hyphal tips; although this is lethal to each tip compartment, the overall activity of the drug is not cidal. Thus, using conidia to address cidity versus static activity is not relevant to the activity of the drug.

Answer: We thank the reviewer for the comments. Unfortunately, technically it is not possible to plate and assess the colony forming units for the caspofungin [0.2 and 0.5 µg/ml, close to minimal effective concentration, (MEC), and supra-MEC] and voriconazole (0.125 and 0.25 µg/ml, sub-inhibitory concentrations) controls since under these concentrations there is extensive conidial germination after 48 h. However, we added microscopy pictures that correspond to the experiments in the Figure 2A showing the fungal growth at caspofungin 0.5 µg/ml and voriconazole 0.25 µg/ml. The main objective of this experiment was to check the cidity of the combination of brilacidin with either caspofungin or voriconazole. The paragraph was modified aiming to make the information clearer (Page 8, lines 219 to 227): “BRI MIC for *A. fumigatus* was measured as higher than 80 µM (Table 1 and Supplementary Table S1) and the combination of 20 µM BRI with

0.2 or 0.5 µg/mL of CAS or 0.125 and 0.25 µg/mL of VOR for 48 h at 37 °C completely inhibited conidial germination while these concentrations of CAS and VOR alone allowed slow conidial germination (Figures 2A and 2B). To test if these conidia were still viable after treatment, we plated them in minimal medium and tested for conidial viability assessing the number of colony forming units. The combination of 0.2 or 0.5 µg/mL of CAS with 20 µM BRI reduced *A. fumigatus* conidial viability by 85% and 100%, respectively (Figure 2A)."

2. In table 1, there are symbols used for "little growth" and "partial inhibition"; it is not clear what little growth means and how it is different from partial inhibition.

Answer: We thank the reviewer for the comment. The nomenclature of the symbols was changed to: "Obs.: CAS=caspofungin; BRI=Brilacidin. CM7555, DPL1033, and MD24053 are clinical isolates resistant to caspofungin. The mechanism of CAS^R in CM7555 is unknown while strains DPL1033 and MD24053 have mutations S679P and F675S in the *fks1* gene. (+): CAS Minimal Effective Concentration (MEC) inhibition; (-): more inhibition than CAS MEC inhibition ; and (---) total inhibition; ND=not determined"

3. Figure 3C does not add much to the figure and could easily be added to supplementary.

Answer: If acceptable to the reviewer and editor, we prefer to keep Figure 3C because this figure demonstrates that the *mpkA* and *calA* genes can complement the mutants, strongly indicating the single genetic modification present in the $\Delta mpkA$ and $\Delta calA$ are the deletion of the genes *mpkA* and *calA*, respectively.

4. There are no methods stated as to how 'fungicidal' activity was assessed for the zygomycetes species tested in Table 2.

Answer: This is now added as a footnote to Table 2: "Fungicidal activity was measured by plating the cells or conidia after 48 hours incubation with the different drug concentrations on solid minimal medium and calculating the colony forming units divided by the initial inoculum; ND=Not determined"

5. Page 10 " These results strongly suggest that CrzA is important for synergistic activity of BRI+CAS" has not really been experimentally addressed. Only the sensitivity to BRI was tested for *calA* and *mpkA* – is the interaction between CAS and BRI still synergistic against these strains? It is clear that the combination treatment activates the CWIP through the nuclear localization of CrzA and therefore the CWIP probably plays a role in response to these treatments.

Answer: We thank the reviewer for the comment and the suggestions. The sentence was changed to (Page 13, lines 362 to 364): "Taken together, these results strongly indicate that *CalA* and *MpkA* are important for BRI activity, and BRI is most likely impacting the *A. fumigatus* CWI pathway."

We have added the FIC values to these strains in the new Figure 4.

6. Page 13 "Thus, unlike VOR and BRI against *A. fumigatus*, it appears that the BRI can potentiate the activity of POSA against Mucorales fungi in vitro" This conclusion is perplexing because the FICI results for both the Mucorales and *A.*

fumigatus are the same. Both interactions yielded additive FICI values between 0.5 and 1. What is the basis for the conclusion that the combination treatment is different between these fungi?

Answer: We apologize about the mistake. The sentence is wrong and was now corrected (Page 14, lines 407 to 409): "Thus, like VOR and BRI against *A. fumigatus*, it appears that BRI can potentiate the activity of POSA against Mucorales fungi *in vitro*."

7. The materials and methods (and Table 1) state "Wild type *A. fumigatus*" was used but then CEA17 is listed – my assumption is this is a mistake and should say CEA10? CEA17 is a uridine and uracil auxotroph; there is no mention of supplementation in the paper and the use of this strain in animals would not be appropriate since it does not grow in mouse lungs.

Answer: We apologize again about this mistake. The name of the strain is wrong. We have used a prototrophic strain $\Delta ku80 pyrG^+$ and this was modified across the manuscript.

8. The *calA* results are contradictory. In the discussion the authors state "Complementary to this screening, we also identified calcineurin to be more sensitive to BRI" however Figure 3D shows the opposite result where the same concentration of BRI has no effect on the *calA* mutant.

Answer: We thank the reviewer for the comment. We removed this sentence and modified the text to (Page 17, lines 509 to 514): "Since $\Delta mpkA$ and $\Delta calA$ have reduced growth when compared to the wild-type strains, these results were confirmed by using chelerythrine and calphostin C (inhibitors of protein kinase C) and cyclosporin (an inhibitor of calcineurin). Complementary to this screening, we also identified increased CrzA:GFP translocation to the nucleus in the presence of BRI+CAS."

9. Similar to the above comment, some of the discussion is contradictory to the results in the paper - "it remains to be investigated how BRI can potentiate VOR" though none of the results of the paper suggest there is any interaction with voriconazole including the FIC checkerboard assays and the inability of BRI to overcome VOR resistance.

Answer: We thank the reviewer for the comment. We modified the sentence to (Page 18, lines 538 to 539): "It remains to be investigated how BRI can bypass CAS-resistance in *A. fumigatus*."

10. It is unclear if the proper statistics have been applied throughout the paper. None of the figures describe what tests were used and what the comparison for each test is except for Figure 6. The Statistical Analysis portion of the Materials and Methods only mentions a t-test which is not appropriate for all the analyses in the paper. Particularly the analysis of Figure 5 which includes sets of data with multiple comparisons and animal data for which a t-test may or may not be appropriate (proper tests should be used to determine whether parametric vs non-parametric is appropriate). In figure 3D the important comparison would be whether the effect of BRI treatment on *mpkA* or *calA* is different from that of WT;

it might be better to present these data as the change in growth (/metabolism) with the addition of BRI to directly compare the effects between strains, given that this mutant strain already has a defect compared to WT. The actual difference in the treatment effect does not seem to be that large (~10%).

Answer: We thank the reviewer for the comments and apologize about the mistakes. In the animal experiments of Figure 5C (now Figure 6C), we have used One-way analysis of variance (ANOVA) followed by Tukey's multiple comparison test. We have added a Table with the statistical comparisons next to the Figure 5C (now Figure 6C). The same analysis was used for Figure 5A and Figure 5B (now Figures 5A and 5B). This is now added to the legend of Figure 5 (now Figure 6).

We thank the reviewer for the suggestion of analysis of the statistics of Figure 3D. We have followed the suggestion and modified the analysis comparing each strain exposed to brilacidin to the corresponding strain without exposure to brilacidin.

Reviewer #3 :

Fernanda Dos Reis et al analyzed the synergy of brilacidin, a mimetic of a host defense peptide and caspofungin. The work is well done and the data are sound. However, the works lacks novelty

Answer: We thank the reviewer for recognizing the importance of our work. However, we respectfully disagree about the fact that our work lacks novelty. We believe our work has a great degree of novelty and therapeutic importance. That is the first time brilacidin is reported as a synergizer of caspofungin against *Aspergillus fumigatus*. Also, converting caspofungin acetate from a static drug to a cidal drug when combined with brilacidin (a drug shown to be safe in clinical trials) has tremendous translational potential in current treatment of drug-resistant fungal infection.

1. The topic is not new and the synergy between antifungals and host derived peptides has been shown previously. Indeed, looking for synergy between antifungal molecules has been recommended for many years and lead to many publications.

Answer: The reviewer is absolutely right about the fact that the synergy between antifungals and host derived peptides has been shown previously. However, we are not working with a host derived peptide but with a host defense peptide mimetic, brilacidin. The concepts are very different. Brilacidin is a fully synthetic polymer-based, non-peptidic small molecule drug candidate exhibiting antimicrobial and immunomodulatory properties, designed to mimic the structure and activity of Human Host Defense Peptides (HDPs or antimicrobial peptides). Thus, it is a small molecule, new chemical entity, created so as to mimic the amphiphilic structure of host defense proteins (HDPs), having one surface with positively charged groups (cationic) and the opposite surface consisting of hydrophobic groups. With this general synthetic form, there is no need for an agent to be of the size or composition of naturally-occurring proteins to effectively function as a HDP, as the ability to act as a HDP is retained by the much smaller synthetic amphiphilic molecule. These HDP mimetic compounds, including Brilacidin, have been shown to exhibit properties similar to those reported for HDPs, notably antibacterial and anti-inflammatory and immune modulatory activities. The fact that this class of arylamides compounds show antifungal activity and can potentiate azoles and caspofungin against several human fungal pathogens brings these compounds as a novel therapeutic option. Due to these differences, HDPs can selectively and variously interact with pathogens, increasing their susceptibility to proteolysis and degradation. It is very difficult to work therapeutically with ordinary host peptides considering their production, documented toxicity, and susceptibility to degradation and commercialization. As a polymer-based, non-peptidic small molecule drug, brilacidin has all the

advantages of stability and commercialization of other currently commercial drugs and has acceptable safety profiles in humans.

2. The increased antifungal efficacy between the inhibition of the CWI pathway (mainly due to specific gene deletions) and echinocandins is well known. Moreover, in these early publications, the role of phosphatases regulating the Hog an CWI pathways has been shown.

Answer: We thank the the reviewer for the comment. The reviewer is likely referring to caspofungin adjuvants such as cyclosporin and Hsp90-inhibitors, that can enhance caspofungin activity. Our data show that brilacidin is a synergizer of caspofungin, potentializing its activity and that the CWI pathway is important in its mechanism of action.

3. The direct activity of brilacidin +/-caspofungin against beta 1,3 glucan synthase activity has not been biochemically investigated.

Answer: We thank the reviewer for the suggestion. Our data shows brilacidin is inhibiting the *A. fumigatus* CWI pathway. MpkA and the phosphatase calcineurin turn on the transcription factors RlmA and CrzA, which translocate to the nucleus and regulate the activation of several stress responses and cell wall modifications, including chitin synthase gene expression and positively impacting the expression of genes involved in β -1,3-glucan and α -1,3-glucan biosynthesis. To measure if brilacidin impacts beta-1,3-glucan synthase activity is an interesting experiment but we believe it will not add much to this current manuscript. Consequently, these studies can be conducted in future studies.

4. The finding that this peptide is synergistically active against all human pathogenic fungi in combination with caspofungin and azoles questions the specificity of the mode of action of this mimetic. Is this synergistic effect due more to the activity of this peptide on membrane depolymerization rather than on the CWI pathway?

Answer: The depolarization of the cell membrane has been reported for brilacidin against some bacterial species. However, the full mechanism of action for brilacidin has not been completely understood. That is the first report about brilacidin in eukaryotic cells like fungi (another novel point of our studies). Depolarization of the cell membrane is not the main mechanism of action against *Aspergillus fumigatus* considering we are showing brilacidin is a synergizer and needs very high concentrations to inhibit the *A. fumigatus* fungal growth. We clearly show in the manuscript the cell wall integrity pathway (CWI) is the main brilacidin target. We would like to bring the reviewer's attention to the fact that agents targeting the CWI pathway potentialize non-inhibitory concentrations of brilacidin. This is further validated by the use of specific mutants in this pathway. Therefore, it is highly unlikely that depolarization of the cell membrane explains these phenotypes since brilacidin alone does not depolarize the *A. fumigatus* cell

membrane, but only brilacidin combined with caspofungin. We believe the cell membrane depolarization is most likely a secondary effect of the disruption of the cell membrane by the intense osmotic pressure caused by caspofungin (inhibiting the beta 1-3 glucan synthase) combined with brilacidin inhibiting the CWI pathway. In addition, the *in vivo* work and the lack of human cell toxicity completely validates its importance as a new panfungal adjuvant that could potentialize caspofungin and azoles activities against several human fungal pathogens.

REVIEWERS' COMMENTS

Reviewer #1 (Remarks to the Author):

The authors have responded to all of my comments and added new experimental data/explanations, the draft is much more readable. I have no further major comments, but would suggest the authors:

1. Provide high-quality figures
2. Add more information about BRI synergism pathway to Fig. 8 model
3. Proof-read the manuscript closely for tense mismatches.

Reviewer #2 (Remarks to the Author):

Though the revised manuscript presented by Fernanda dos Reis et al., provides some improved data to support the role of the CWI in the fungal response to the host peptide mimetic BRI, the manuscript still draws conclusions that are not supported by the data. As it stands, the authors have not adequately addressed reviewer concerns and I would not recommend this manuscript for publication in Nature Communications.

The experiment in Figure 2A is still an inappropriate way to assess fungicidal versus static activity. The activity of CAS on conidia is not relevant and therefore this experiment does not reflect the activity of these two drugs as they would be used clinically (to treat hyphal cells/biofilms). It is very technically difficult to distinguish between the two modes of activity in filamentous fungi, especially when one compound interacts with the membrane and can interfere with live/dead staining. However, the claim that the activity has been converted from fungicidal to fungistatic for *A. fumigatus* is still unsubstantiated. Given that this assay is irrelevant to the drug activity and the proper controls are not technically possible, another assay should be used to evaluate this. Otherwise, using the terms fungicidal and fungistatic might not be appropriate here and the interaction should be described as synergy.

In the authors response to reviewer 3 they state, "Depolarization is not the mechanism of action against *A. fumigatus*...we clearly show in the manuscript that the CWIP is the main brilicidin target". These conclusions here and in the text are overstatements of the data and have not been sufficiently tested. The membrane effect is only tested at a concentration of BRI that has no effect on *A. fumigatus* growth.

DiBAC staining should be performed at concentrations that have antifungal activity or ideally over a range of drug concentrations from sub- to supra- MIC. Though the data support a role for the CWIP in response to BRI and CAS treatment (to be expected since this has already been demonstrated for CAS) the CWIP as the target of BRI is not supported. BRI and CAS treatment both seem to activate the CWIP through the activation of chitinase genes (similar activation of some genes in response to both CAS or BRI alone). If BRI was inhibiting the CWIP, I would expect to see a reduction in chitinase gene expression in CAS and BRI compared to CAS alone. Similarly, the addition of BRI only enhances the translocation of CrzA in response to CAS rather than inhibit the activation by CAS. Overall, the data support the activation of the CWIP in response to both CAS alone and CAS and BRI together. Western blot data of MpkA phosphorylation or the phosphorylation of other CWIP components would be a more direct way to look at inhibition or activation of the CWIP by BRI. Even the combination treatments of BRI with the PKIs and other chemical inhibitors can be confounded by increased membrane permeability due to BRI treatment (membrane permeability was only tested at much lower concentrations than used in the combination experiments).

Line 501 “We investigated how BRI can converts CAS into a fungicidal drug by” The result showing that caspofungin is fungistatic against WT, mpkA and calA argues that even in the absence of a functioning CWIP, the activity of CAS is still fungistatic. This argues against the authors’ conclusion that inhibition of the CWIP by BRI converts CAS to a fungicidal drug.

Line 165 - The terms synergizers and enhancers should be defined in the text; not all studies define these terms the same way and making the reader look up the definition is cumbersome

The images in Figure 2B are poor quality and therefore it is difficult to see what the treated hyphae look like morphologically.

ANSWERS TO THE REVIEWERS

Reviewer #1:

The authors have responded to all of my comments and added new experimental data/explanations, the draft is much more readable. I have no further major comments, but would suggest the authors:

Answer: We thank the reviewer for the suggestions and for the continuous improvement of the quality of our manuscript.

1. Provide high-quality figures

Answer: We have now provide high-quality figures.

2. Add more information about BRI synergism pathway to Fig. 8 model

Answer: Figure 8 was removed following editorial requests.

3. Proof-read the manuscript closely for tense mismatches.

Answer: We have proof-read the manuscript for tense mismatches.

Reviewer #2:

Though the revised manuscript presented by Fernanda dos Reis et al., provides some improved data to support the role of the CWI in the fungal response to the host peptide mimetic BRI, the manuscript still draws conclusions that are not supported by the data. As it stands, the authors have not adequately addressed reviewer concerns and I would not recommend this manuscript for publication in Nature Communications.

Answer: We thank the reviewer for the further comments and emphasizing some concepts. We have now replaced the idea of BRI converting CAS into a fungicidal drug and attenuated some of our conclusions.

The experiment in Figure 2A is still an inappropriate way to assess fungicidal versus static activity. The activity of CAS on conidia is not relevant and therefore this experiment does not reflect the activity of these two drugs as they would be used clinically (to treat hyphal cells/biofilms). It is very technically difficult to distinguish between the two modes of activity in filamentous fungi, especially when one compound interacts with the membrane and can interfere with live/dead staining. However, the claim that the activity has been converted from fungicidal to fungistatic for *A. fumigatus* is still unsubstantiated. Given that this assay is irrelevant to the drug activity and the proper controls are not technically possible, another assay should be used to evaluate this. Otherwise, using the terms fungicidal and fungistatic might not be appropriate here and the interaction should be described as synergy.

Answer: We thank the reviewer by driving again our attention for the difficulties imposed by using this idea of BRI as a compound that converts CAS from a fungistatic into a fungicidal drug. Based on this relevant comment, we have now removed all the sentences that suggest that brilacidin is converting caspofungin from a fungistatic drug into a fungicidal drug and replace this by the word synergism, as suggested by the reviewer.

In the authors response to reviewer 3 they state, "Depolarization is not the mechanism of action against *A. fumigatus*...we clearly show in the manuscript that the CWIP is the main brilacidin target". These conclusions here and in the text are overstatements of the data and have not been sufficiently tested. The membrane effect is only tested at a concentration of BRI that has no effect on *A. fumigatus* growth. DiBAC staining should be performed at concentrations that have antifungal activity or ideally over a range of drug concentrations from sub- to supra- MIC.

Answer: We thank the reviewer for the comment. We have now tested the depolarization of the membrane by using exactly the same concentrations that show synergism, i.e., 20 μ M of BRI and 0.125 μ g of CAS. As it can be seen in the new Figure 2A, these concentrations of BRI or CAS alone show about 10- to 3-fold less depolarization than when BRI+CAS are applied together. These results once more emphasize that the synergism of BRI+CAS is not only due to the membrane depolarization but to a combination of both misexpression of the cell

wall integrity pathway and membrane depolarization. We added these ideas to the Results section (Page 9, lines 239 to 248): “Untreated *A. fumigatus* had 0% fluorescent germlings while exposure to either 0.125 µg/mL CAS, 20 µM BRI, or a combination of both resulted in ~25%, 10%, and 80% fluorescent germlings, respectively (Figure 2E). Germlings previously transferred to (minimal medium) MM without glucose (non-carbon source, NCS) for 4 hours before adding CAS, BRI, or CAS+BRI for 10 min and subsequently DIBAC₄(3) have not shown any fluorescent difference with the cells that were grown in the presence of the carbon source glucose (Figure 2E). These results suggest that BRI as well as CAS is passively transported through the cell membrane without the need for ATP.”

And to the Discussion section (Pages 17 and 18, lines 502 to 520 and 528, respectively): “We investigated the mechanism of action for BRI and CAS synergism by a combination of screening PKIs and a collection of *A. fumigatus* null mutants for the catalytic subunits of phosphatases. We observed two PKI compounds, a p21-Activated Kinase Inhibitor FRAX486 and a STK25 inhibitor PP121, both members of the sterile 20 superfamily of kinases, as potentiating the BRI activity against *A. fumigatus*. Since p21 activated kinases (PAKs) are involved in signaling through MAPK pathways⁶¹, we tested if other *A. fumigatus* MAPK are more sensitive to BRI and identified MpkA as being so, the apical MAPK that is regulated by protein kinase C and coordinates the CWI pathway⁸⁶. Since $\Delta mpkA$ and $\Delta calA$ have reduced growth when compared to the wild-type strains, these results were confirmed by using chelerythrine and calphostin C (inhibitors of protein kinase C) and cyclosporin (an inhibitor of calcineurin). Complementary to this screening, we also identified increased CrzA:GFP translocation to the nucleus in the presence of BRI+CAS. We demonstrated that the combination of BRI+CAS affects several aspects of the CWI pathway, such as altered increased or decreased expression of chitin synthase genes, and the exposure of chitin to the cell wall, strongly suggesting that BRI is inhibiting the *A. fumigatus* CWI pathway (Figure 8). Thus, we propose that BRI+CAS has a synergism that is able to affect *A. fumigatus* viability through multiple mechanisms of action, encompassing functional changes/depolarization of the microorganism cell membrane, interference to calcineurin signaling (as for instance the increased CrzA translocation), and misexpression of the cell wall integrity pathway, and necessary compensatory reactions (since CAS is also present and prevents further β -1,3-glucan biosynthesis). Consequently, cell wall assembly and remodelling, which prevents resistance against osmotic forces, is inhibited which then leads to cell lysis.”

Though the data support a role for the CWIP in response to BRI and CAS treatment (to be expected since this has already been demonstrated for CAS) the CWIP as the target of BRI is not supported. BRI and CAS treatment both seem to activate the CWIP through the activation of chitinase genes (similar activation of some genes in response to both CAS or BRI alone). If BRI was inhibiting the CWIP, I would expect to see a reduction in chitinase gene expression in CAS and BRI compared to CAS alone. Similarly, the addition of BRI only enhances the translocation of CrzA in response to CAS rather than inhibit the activation by CAS. Overall, the data support the activation of the CWIP in response to both CAS alone and CAS and BRI together. Western blot data of MpkA phosphorylation or the phosphorylation of other CWIP components would be a more direct way to look at inhibition or activation of the CWIP by BRI.

Answer: We would like to thank the reviewer for the comments. I think the reviewer is right in the sense the combination of BRI+CAS is not always inhibiting every single chitin synthase induction or inhibiting the translocation of CrzA to the nucleus. For some of the chitin synthases, the combination of the drugs is either inhibiting or increasing the expression. The increased expression of the cell wall integrity pathway is also deleterious to the cell, impacting the cell wall organization, and affecting its functionality. Considering the reviewer suggestions, we think the best concept to define what is happening with the cell wall integrity pathway due to its heterogeneous nature is rather a “misexpression” than an “inhibition”. Accordingly, we replaced the idea of inhibition by misexpression along the manuscript. Please, see response to the previous question and how we modified the Discussion section to address some of the questions raised by the reviewer.

Even the combination treatments of BRI with the PKIs and other chemical inhibitors can be confounded by increased membrane permeability due to BRI treatment (membrane permeability was only tested at much lower concentrations than used in the combination experiments).

Answer: Unfortunately, the reviewer is not right about the concentration that was used in these experiments. All the combination experiments performed with PKIs, and with the specific inhibitors for calcineurin and protein kinase C, were done at the concentration of 20 μ M of BRI, a sub-inhibitory concentration in terms of membrane depolarization for *A. fumigatus*. As stated above, we have now tested 20 μ M in the depolarization experiments.

Line 501 “We investigated how BRI can converts CAS into a fungicidal drug by” The result showing that caspofungin is fungistatic against WT, *mpkA* and *calA* argues that even in the absence of a functioning CWIP, the activity of CAS is still fungistatic. This argues against the authors’ conclusion that inhibition of the CWIP by BRI converts CAS to a fungicidal drug.

Answer: We have now removed all the sentences that suggest that brilacidin is converting caspofungin from a fungistatic drug into a fungicidal drug and replace this interaction by the word synergism, as suggested by the reviewer (Page 17, lines 502 to 505): “We investigated the mechanism of action for BRI and CAS synergism by a combination of screening PKIs and a collection of *A. fumigatus* null mutants for the catalytic subunits of phosphatases”.

We also thank the reviewer for this observation about the fact that activity of CAS against the mutants $\Delta calA$ and $\Delta mpkA$ is still fungistatic. This is true but these two mutants are more sensitive to CAS and when these mutants are exposed to 0.25 μ g/ml of CAS (the MEC for the wild-type), CAS becomes fungicidal against these strains. We have complemented the sentence, stating the MECs for the mutants (Page 12, lines 346 to 349): “There is an increased susceptibility of $\Delta mpkA$ and $\Delta calA$ to lower combinations of BRI+CAS when compared to the wild-type strain (MECs for $\Delta calA$ and $\Delta mpkA$ are 0.125 μ g/ml and 0.06 μ g/ml, respectively, Figures 4A and 4B)”.

Line 165 - The terms synergizers and enhancers should be defined in the text; not all studies define these terms the same way and making the reader look up the definition is cumbersome

Answer: We have already defined enhancers and synergizers in the previous sentence (Pages 6 and 7, lines 161 to 163 and 164 to 166, respectively): “Based on this assay we defined enhancers as the compounds that alone could inhibit over 30 % of *A. fumigatus* metabolic activity but in combination with CAS inhibited even more, while synergizers were defined as compounds which alone inhibited less than 30 % of the fungal metabolic activity but in combination with CAS inhibited more than 30 % (for further definitions and applications of synergizers and enhancers, see ³⁸⁻⁴¹).

The images in Figure 2B are poor quality and therefore it is difficult to see what the treated hyphae look like morphologically.

Answer: The images of the Figure 2B were improved.